

# The one-dimensional Bose gas with strong two-body losses: the effect of the harmonic confinement

**Lorenzo Rosso[1*], Alberto Biella[1,2] and Leonardo Mazza[1]**

**1** Université Paris-Saclay, CNRS, LPTMS, 91405 Orsay, France
**2** INO-CNR BEC Center and Dipartimento di Fisica, Università di Trento, 38123 Povo, Italy

★ lorenzo.rosso@universite-paris-saclay.fr

## Abstract

We study the dynamics of a one-dimensional Bose gas in presence of strong two-body losses. In this dissipative quantum Zeno regime, the gas fermionises and its dynamics can be described with a simple set of rate equations. Employing the local density approximation and a Boltzmann-like dynamical equation, the description is extended to take into account an external potential. We show that in the absence of confinement the population is depleted in an anomalous way and that the gas behaves as a low-temperature classical gas. The harmonic confinement accelerates the depopulation of the gas and introduces a novel decay regime, which we thoroughly characterise.

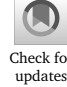
# 1   Introduction

The study of quantum many-body physics with ultra-cold gases has recently attracted considerable attention [1]. The unprecedented possibility of studying a quantum many-body system in isolated conditions and with negligible effects from the environment has made possible the experimental characterisation of its closed-system dynamics in well-controlled settings, see for instance the experiments reported in Refs. [2–4]. Restricting our focus to one-dimensional Bose gases that at low temperature are described by the Lieb-Liniger model [5,6], several developments, among which the generalised hydrodynamics [7,8], have produced a theoretical framework that successfully describes the experiments where bosonic gases are put out of equilibrium via a quantum quench of the external potential [9–11].

Yet, even these quantum systems cannot be considered as perfectly isolated, and the most relevant form of environment to which they are coupled is represented by the surrounding vacuum into which atoms or molecules can leak [12–14]. Even if current experiments do not suffer from significant loss effects, the community has recently witnessed a renewed interest in the dynamics that they induce in correlated quantum gases. On the one hand, they need to be taken into account in order to develop a quantitative description of experiments [9,12–14]. On the other hand, it was early recognised that open-system dynamics (such as losses) can be extremely interesting and can lead to the appearance of phenomena where quantum mechanics and quantum coherence play an important role [15–20]. One well-explored phenomenon is that of cooling by atom losses [21–24]. Moreover, the recent possibility of inducing and engineering losses in well-controlled conditions has finally opened the path to the experimental study and quantum simulation of many-body dissipative dynamics [25].

In this article we focus on the one-dimensional Bose gas with two-body losses. From an experimental viewpoint, our work has several motivations. First, the recent advances in the cooling to quantum degeneracy of novel atomic species have produced ultra-cold gases characterised by strong inelastic interactions. For instance, ytterbium atoms are characterised by an excited and long-lived metastable state that suffers from two-body recombination processes, so that a quantum-degenerate gas in this metastable state is a natural setup to study the interplay between two-body losses and quantum dynamics [26–28]. Second, photoassociation processes can induce two-body losses with controllable strength in standard alkaline atoms, and have already been used in several contexts, such as in the study of dissipative phase transition [25]. Last, the use of molecules gives the possibility of studying a quantum degenerate gas subject to two-body losses and it is one of the setups where these experimental studies

were first performed [29, 30]. For completeness we mention that experiments exist also in the fermionic case [31–33].

From the theoretical viewpoint, a first important step in the description of the strongly-correlated and lossy Bose gas has been presented in Refs. [34, 35], focusing primarily on the limit of weak losses (the case of strong interactions and fermionisation is discussed only for one-body losses). Here, instead, we are interested in the case of strong two-body losses, and our goal is to characterise the simplest experimental observable, namely the dynamics of the total number of particles composing the gas. This problem has already been discussed for a lattice gas described by the Bose-Hubbard model without any external confinement in Ref. [36]. There, it is shown that the gas effectively fermionises, namely that it becomes a gas of hard-core bosons, thus developing strong particle-particle correlations and a peculiar momentum distribution function that does not resemble any equilibrium situation. Concerning the continuum situation, in Ref. [37] the authors present a solution of the dissipative Lieb-Liniger model, where the interaction constant has also an imaginary part. The study, that has the merit of highlighting in a transparent way the fermionisation induced by strong two-body losses, does not produce any prediction for the dynamics of the population of the gas. Finally, we mention the existence of a conspicuous literature focusing on the problem of two-body losses in one-dimensional fermionic gases [38–41].

In this article we present a theoretical description of the effect of a harmonic confinement on the dynamics of the one-dimensional Bose gas with strong two-body losses. As a starting point, and for pedagogical reasons, we show that in the absence of harmonic confinement the long-time dynamics deviates from naive mean-field expectations; in particular we show that the gas is turned into a low-temperature classical gas described by a Maxwell-Boltzmann momentum distribution function. The harmonic confinement has a qualitative impact on such behaviour. Two regimes are identified: that of weak confinement, where the period of an oscillation is long with respect to the loss rate, and of strong confinement, with opposite properties. Whereas in the former case the dynamics has all qualitative features of the homogeneous system, in the latter case we find that the harmonic trap accelerates the depletion of the gas. The methodology that we employ relies on the fact that strong two-body losses fermionise the gas; the dynamical properties are indeed obtained by characterising the population of the emergent fermionic momenta (or rapidities) using a Boltzmann-like equation.

The article is organised as follows. In Sec. 2 we introduce the theoretical model, we define the limit of strong two-body losses and present the set of rate equations that describe its dynamics in the absence of any external potential. In Sec. 3 we solve the aforementioned equations assuming an equilibrium initial configuration. In Sec. 4 we consider the presence of the harmonic confinement: we generalise the equations describing the homogeneous gas and discuss the dynamics of an initial equilibrium state. Our conclusions are presented in Sec. 5. Five appendices conclude the article.

## 2 Homogeneous Bose gas with strong two-body losses: the Quantum Zeno effect

We consider a bosonic gas trapped in one dimension and subject to strong two-body losses; for simplicity, we consider a homogeneous sample of length $L$ with periodic boundary conditions. We introduce a field $\Psi(x)$ satisfying canonical commutation relations $[\Psi(x), \Psi(x')] = 0$ and $[\Psi(x), \Psi^\dagger(x')] = \delta(x - x')$, and describe the dynamics using the Lindblad master equation,

which make sense only in 1D (see Ref. [20]), $\frac{\partial}{\partial t}\rho(t) = -\frac{i}{\hbar}[H,\rho(t)] + \mathcal{D}[\rho(t)]$ with:

$$H = H_{\text{kin}} + H_{\text{int}} = \int \Psi^\dagger(x)\left(-\frac{\hbar^2}{2m}\partial_x^2\right)\Psi(x)\mathrm{d}x + \frac{g}{2}\int \Psi^{\dagger 2}(x)\Psi^2(x)\mathrm{d}x; \qquad (1a)$$

$$\mathcal{D}[\rho] = \frac{\gamma}{2}\int\left(\Psi^2(x)\rho\Psi^{\dagger 2}(x) - \frac{1}{2}\left\{\Psi^{\dagger 2}(x)\Psi^2(x),\rho\right\}\right)\mathrm{d}x. \qquad (1b)$$

In Eq. (1a) we recognize the Lieb-Liniger model; the term in Eq. (1b) accounts for two-body losses. A study of this model in the weakly-dissipative limit has been presented in Ref. [34, 35]; an ab-initio derivation of the master equation and critical discussion of the underlying approximations can be found in Ref. [37].

## 2.1 Strong dissipation and fermionisation

We are interested in studying this model in the strongly-dissipative regime, which is often called quantum Zeno regime. In order to better clarify this limit, it is useful to consider the non-Hermitian Hamiltonian associated to the master equation; we thus introduce the non-Hermitian version of the Lieb-Liniger model that has been thoroughly discussed in Ref. [37]:

$$H_{\text{NHLL}} = \int \Psi^\dagger(x)\left(-\frac{\hbar^2}{2m}\partial_x^2\right)\Psi(x)\mathrm{d}x + \frac{g + i\hbar\gamma}{2}\int \Psi^{\dagger 2}(x)\Psi^2(x)\mathrm{d}x. \qquad (2)$$

The strength of the non-linear term which accounts for the two-body elastic and inelastic interaction, is quantified by the dimensionless parameter:

$$\xi = \frac{m(g + i\hbar\gamma)}{\hbar^2 n}, \qquad (3)$$

where $n$ is the one-dimensional density of the gas ($H_{\text{NHLL}}$ is number-conserving and thus the density $n = N/L$ is well-defined). According to Ref. [37], the model can be mapped to a Tonks-Girardeau gas whenever $|\xi| \to \infty$. It is well-known that the Bose gas becomes a gas of hard-core particles when $g \gg \hbar^2 n/m$; this is true also when $g = 0$ and the following inequality is satisfied:

$$\gamma \gg \frac{\hbar n}{m}. \qquad (4)$$

This inequality defines the regime of strong dissipation. If we consider a $^{87}$Rb gas (atomic mass: $m = 1.43 \times 10^{-25}$ Kg) with typical density $n \simeq 1 \times 10^7$ m$^{-1}$ as in Ref. [9], we obtain that in order to enter the strongly-dissipative regime, the inequality reads: $\gamma \gg 7.24 \times 10^{-3}$ m·s$^{-1}$.

In this regime the bosonic model is mapped to non-interacting fermions by the Jordan-Wigner transformation $c(x) = (-1)^{N_{[0,x]}}\Psi(x)$, where $N_{[0,x]}$ is the number of bosons in the interval $[0,x]$. The new field operators satisfy canonical anticommutation relations: $\{c(x),c(x')\} = 0$ and $\{c(x),c^\dagger(x')\} = \delta(x - x')$ and the transformed Hamiltonian is a free-fermion model:

$$H_{\text{NHLL}} = \int c^\dagger(x)\left(-\frac{\hbar^2}{2m}\partial_x^2\right)c(x)\mathrm{d}x. \qquad (5)$$

Although the dissipation does not conserve the number of particles, it conserves the parity of the number of particles. Thus, in order to study the problem in the sector with an even number of particles, we introduce the field operators in momentum space via Fourier transform: $c_k = L^{-\frac{1}{2}}\int c(x)e^{-ikx}\mathrm{d}x$ where $k = \pi(2n + 1)/L$ and $n \in \mathbb{Z}$. The Hamiltonian in momentum space reads:

$$H_{\text{NHLL}} = \sum_k \frac{\hbar^2 k^2}{2m}c_k^\dagger c_k. \qquad (6)$$

Summarizing, in the limit of infinitely-strong dissipation $|\xi| \to \infty$, Hamiltonian (2) is well-described by a non-interacting fermionic model, whose eigenstates and eigenenergies are defined by the occupation numbers of the modes $k$.

This information is of great help also for discussing the stable modes of the master equation (1). Let us for instance consider a density matrix that is diagonal in the basis of the modes $k$ and that is solely parametrised by a set of $\{\lambda_k\}$ coefficients related to the occupation of the modes:

$$\rho = \prod_k \frac{e^{-\lambda_k c_k^\dagger c_k}}{1 + e^{-\lambda_k}} = \prod_k \frac{1 + (e^{-\lambda_k} - 1)c_k^\dagger c_k}{1 + e^{-\lambda_k}}. \tag{7}$$

In order to prove that the latter density matrix is a stationary state of the master equation (1) we recast Eqs. (1) as:

$$\frac{\partial}{\partial t}\rho(t) = -\frac{i}{\hbar}\left(H_{\mathrm{NHLL}}\rho - \rho H_{\mathrm{NHLL}}^\dagger\right) + \frac{\gamma}{2}\int \Psi(x)^2 \rho \Psi(x)^{\dagger 2} dx. \tag{8}$$

Within the fermionised approximation, $H_{\mathrm{NHLL}}$ is given by Eq. (6), which commutes with the density matrix (7); we consequently conclude that the first part of the r.h.s. of Eq. (8) vanishes. The part proportional to $\gamma$ vanishes as well because $\Psi(x)^2$ is equal to zero for a fermionised state. We thus conclude that the fermionic modes identified above with the non-Hermitian Lieb-Liniger model are stable modes of the master equation (1).

The operators $\{c_k^\dagger c_k\}_k$ constitute an infinite set of operators that commute with the Hamiltonian; the state (7) is the associated generalised Gibbs ensemble (GGE) [42]. The fermionic momenta, also called rapidities in the Bethe-ansatz context, can be observed in experiments by letting the gas expand in the one-dimensional tube [10, 43–46]. This is different from performing a standard time-of-flight experiment on a one-dimensional gas: in this case the gas expands freely in three-dimensional space, and one instead probes the bosonic momentum distribution function [1].

## 2.2 Rate equations for the dissipative dynamics

In reality, in any experimental setting, the parameter $\xi$ is never going to have infinite modulus. The main consequence of this fact is that the fermionic excitations described by the $c_k$ acquire a finite decay time and become quasi-stable. From an experimental viewpoint, this means that particles leak from the sample and the setup is depleted. It is interesting to observe that if the initial density satisfies the inequality (4), the gas will remain in the strongly-dissipative regime at all times because $n(t)$ is a non-increasing function (the system can only lose particles). The curious thing about the system that we are studying is that even if we are working in the strongly-dissipative regime, we have identified a set of modes (or quasiparticles) which are almost stable. The dynamics of these modes is thus weakly dissipative: as we will see, their decay rate scales as $1/\gamma$ and this counterintuitive effect is known in the literature as environment-induced quantum Zeno effect [15, 47].

It has already been observed that in the presence of weak losses, the dynamics can be described with a time-dependent state that is a simple generalisation of the GGE proposed in (7) (see Refs. [36, 48, 49]):

$$\rho(t) = \prod_k \frac{e^{-\lambda_k(t)c_k^\dagger c_k}}{1 + e^{-\lambda_k(t)}}. \tag{9}$$

Ref. [48] presents the equations that determine the dynamics of the $\lambda_k(t)$ in general. However, there is a one-to-one correspondence between $\lambda_k(t)$ and $n_k(t) \doteq \mathrm{tr}[\rho(t)c_k^\dagger c_k]$:

$$n_k(t) = \frac{1}{1 + e^{\lambda_k(t)}}; \tag{10}$$

in this work, instead of working with $\lambda_k(t)$ we prefer to work with $n_k(t)$.

The first result of this article is the identification of the differential equations obeyed by the $n_k(t)$ for this specific problem:

$$\frac{\partial}{\partial t} n_k(t) = -\Gamma_{\text{eff}} \int_{-\infty}^{+\infty} (k-q)^2 n_k(t) n_q(t) \mathrm{d}q; \qquad \Gamma_{\text{eff}} = \frac{2\hbar^3}{\pi m^2} \frac{\frac{\hbar\gamma}{2}}{g^2 + \left(\frac{\hbar\gamma}{2}\right)^2}. \tag{11}$$

As anticipated, $\Gamma_{\text{eff}}$, which dictates the typical decay rate of the fermionic modes, scales as $\gamma^{-1}$. Considering again the abovementioned $^{87}\text{Rb}$ gas discussed in Ref. [9], we estimate $g \sim 8.9 \times 10^{-36}$ kg·m$^3$·s$^{-2}$ and take a value for $\gamma$ that is in the strongly-dissipative regime, $\gamma = 2 \times 10^{-2}$ m·s$^{-1}$. We obtain: $\Gamma_{\text{eff}} = 4.58 \times 10^{-19}$ m$^3$·s$^{-1}$.

The rate equations (11) generalise those already obtained for a one-dimensional bosonic gas trapped in a lattice and described by the Bose-Hubbard model [36]. The derivation of the equations is given here below and it is obtained by regularising the problem onto a lattice. The uninterested reader can jump directly to Sec. 3 without compromising the understanding of the rest of the article.

## 2.3 Derivation of the rate equations

In order to derive the rate equations (11), we regularise the original problem introducing a short-length cutoff, $a$, that is the shortest length-scale of the problem. In this way, the original master equation is turned into a lattice problem. After introducing the lattice bosonic operators $b_m^\dagger$, the master equation reads: $\frac{\partial}{\partial t}\rho(t) = -\frac{i}{\hbar}[H_L, \rho(t)] + \mathcal{D}_L[\rho(t)]$ with

$$H_L = -J\sum_m \left(b_m^\dagger b_{m+1} + H.c.\right) + \frac{U}{2}\sum_m n_m(n_m - 1); \quad \mathcal{D}_L[\rho] = \gamma_L \sum_m b_m^2 \rho b_m^{\dagger 2} - \frac{1}{2}\left\{b_m^{\dagger 2} b_m^2, \rho\right\}. \tag{12}$$

The relation between the original parameters and the lattice ones is given by: $Ja^2 \leftrightarrow \hbar^2/(2m)$; $Ua \leftrightarrow g$, and $\gamma_L a \leftrightarrow \gamma$. Concerning the relation between the physical length of the original system $L$ and the number of lattice points $M$ it holds that $Ma \leftrightarrow L$.

Following the methods employed in Ref. [36], we fermionise the problem, introduce the momenta $k = \frac{2\pi}{L}(n+1)$, with $n = 1, 2, \ldots M$ and obtain the following rate equations for the occupation number of each fermionic mode:

$$\frac{\partial}{\partial t} n_k(t) = -\frac{4\Gamma_L}{M} \sum_q [\sin(ka) - \sin(qa)]^2 n_k(t) n_q(t), \qquad \Gamma_L = \frac{2J^2\gamma_L}{U^2 + \left(\frac{\hbar\gamma_L}{2}\right)^2}. \tag{13}$$

We now take the limit $a \to 0^+$ and obtain the rate equations that describe our system in the continuum limit. First, we observe that:

$$\Gamma_L a^3 \leftrightarrow \frac{\hbar^3}{m^2} \frac{\frac{\hbar\gamma}{2}}{g^2 + \left(\frac{\hbar\gamma}{2}\right)^2}. \tag{14}$$

Since $L$ is fixed and $a$ is going to zero, the number of lattice points $M$ is increasing; when $M \gg 1$ we can change the sum into an integral via $\sum_k \to \frac{L}{2\pi}\int \mathrm{d}k$ and using $L/M = a$ we obtain:

$$\frac{\partial}{\partial t} n_k(t) = -\frac{4\Gamma_L a}{2\pi} \int_{-\pi/a}^{\pi/a} [\sin(ka) - \sin(qa)]^2 n_k(t) n_q(t) \mathrm{d}q. \tag{15}$$

Let us now take the limit $a \ll k_{\text{max}}^{-1}$, where $k_{\text{max}}$ is the maximal wavevector that is going to be occupied during the dynamics. Since each function $n_k(t)$ is monotously decreasing (its

derivative in Eq. (13) is never positive), it is enough to consider the largest wavevector that is occupied at time $t = 0$; this quantity is well defined. In this limit: $\sin(ka) \sim ka$ for all relevant $k$; moreover the integration limits can be safely expanded to $\pm\infty$. We obtain:

$$\frac{\partial}{\partial t} n_k(t) = -\Gamma_{\text{eff}} \int_{-\infty}^{+\infty} (k-q)^2 n_k(t) n_q(t) \mathrm{d}q; \qquad \frac{2a^3 \Gamma_L}{\pi} \longleftrightarrow \Gamma_{\text{eff}}. \tag{16}$$

Note that $\Gamma_{\text{eff}}$ has a well-defined continuum limit and the good dimensions of $L^3 \times T^{-1}$, since $n_k$ is an adimensional quantity. This concludes our derivation.

# 3 Depletion of an initial equilibrium state

We discuss the depletion dynamics of a Bose gas prepared in the ground state of the Tonks-Girardeau gas ($|\xi| \to \infty$) with density $n_{\text{in}}$. We consider the rate equations (11) with the following initial conditions: $n_k(0) = 1$ for $|k| < \pi n_{\text{in}}$ and $n_k(0) = 0$ otherwise. The initial fermionic momentum profile, $n_k(0)$, is symmetric with respect to $k \to -k$ transformations, and thus $\int (k-q)^2 n_q \mathrm{d}q = \int (k^2 + q^2) n_q \mathrm{d}q$. Since the master equation is invariant under $k \to -k$ exchanges, this property is preserved during the whole dynamics and thus Eq. (11) is substituted by:

$$\frac{\partial}{\partial t} n_k(t) = -\Gamma_{\text{eff}} \int_{-\infty}^{+\infty} (k^2 + q^2) n_k(t) n_q(t) \mathrm{d}q. \tag{17}$$

In order to get a better understanding of the problem we introduce the rescaled momentum $\tilde{k} = k/n_{\text{in}}$ and the rescaled time $\tilde{t} = n_{\text{in}}^3 \Gamma_{\text{eff}} t$ so that the equations read:

$$\frac{\partial}{\partial \tilde{t}} n_{\tilde{k}}(\tilde{t}) = - \int_{-\pi}^{+\pi} (\tilde{k}^2 + \tilde{q}^2) n_{\tilde{k}}(\tilde{t}) n_{\tilde{q}}(\tilde{t}) \mathrm{d}\tilde{q}; \qquad n_{\tilde{k}}(0) = \begin{cases} 1 & \text{for } \tilde{k} \in [-\pi, \pi]; \\ 0 & \text{otherwise.} \end{cases} \tag{18}$$

The adimensional normalised density of the gas reads $\tilde{n}(\tilde{t}) = \frac{1}{2\pi} \int n_{\tilde{k}}(\tilde{t}) \mathrm{d}\tilde{k}$ and it satisfies $\tilde{n}(0) = 1$. It is linked to the physical density via the simple relation $n = n_{\text{in}} \times \tilde{n}$.

## 3.1 "Heating" to a low-temperature classical gas

With the aim of studying the dynamics of the gas, we employ Eq. (18) in order to write the two following equations (see App. A for details on the derivation):

$$\frac{\partial}{\partial \tilde{t}} \tilde{n}(\tilde{t}) = -2 \int_{-\pi}^{+\pi} \tilde{q}^2 n_{\tilde{q}}(\tilde{t}) \mathrm{d}\tilde{q} \times \tilde{n}(\tilde{t}); \qquad \frac{\partial}{\partial \tilde{t}} n_{\tilde{k}}(\tilde{t}) = +\frac{n_{\tilde{k}}(\tilde{t})}{2\tilde{n}(\tilde{t})} \frac{\partial \tilde{n}(\tilde{t})}{\partial \tilde{t}} - 2\pi \tilde{k}^2 \tilde{n}(\tilde{t}) n_{\tilde{k}}(\tilde{t}). \tag{19}$$

If we divide the latter equation by $n_{\tilde{k}}(\tilde{t})$ we obtain the following relation:

$$\int_1^{n_{\tilde{k}}(\tilde{t})} \frac{1}{n'_{\tilde{k}}} \mathrm{d}n'_{\tilde{k}} = +\frac{1}{2} \int_1^{\tilde{n}(\tilde{t})} \frac{1}{\tilde{n}'} \mathrm{d}\tilde{n}' - 2\pi \tilde{k}^2 \int_0^{\tilde{t}} \tilde{n}(\tilde{t}') \mathrm{d}\tilde{t}', \tag{20}$$

which is solved by (see again App. A for the details):

$$n_{\tilde{k}}(\tilde{t}) = \begin{cases} \sqrt{\tilde{n}(\tilde{t})} e^{-2\pi \tilde{k}^2 \int_0^{\tilde{t}} \tilde{n}(\tilde{t}') \mathrm{d}\tilde{t}'}, & \tilde{k} \in [-\pi, \pi]; \\ 0, & \text{otherwise.} \end{cases} \tag{21}$$

This latter relation shows that $n_{\tilde{k}}(\tilde{t})$ is fully determined by $\tilde{n}(\tilde{t})$, and in particular that momenta distribute according to a Gaussian function centred at $\tilde{k} = 0$ and truncated at $\tilde{k} = \pm\pi$.

The variance of the Gaussian is $\left(4\pi\int_0^{\tilde{t}}\tilde{n}(\tilde{t}')d\tilde{t}'\right)^{-1}$, which decays to zero as $\tilde{t}^{-\frac{1}{2}}$ in the long-time limit (see below Sec. 3.2). Thus, at sufficiently long times, we can disregard the truncation at $\tilde{k}=\pm\pi$ and interpret the distribution as a Maxwell-Boltzmann classical distribution $n_{\tilde{k}}=e^{\frac{\mu}{k_BT}}e^{-\frac{\hbar^2\tilde{k}^2}{2mk_BT}}$ with time-dependent temperature and chemical potential. Reintroducing for clarity the dimensional units, they read:

$$T(t)=\frac{\hbar^2}{4\pi\Gamma_{\text{eff}}k_Bm}\frac{1}{\int_0^t n(t')dt'}\sim\frac{1}{t^{\frac{1}{2}}};\qquad \mu(T)=\frac{k_BT(t)}{2}\ln\left(\frac{n(t)}{n_{\text{in}}}\right)\sim\frac{1}{t^{\frac{1}{2}}}\ln t. \tag{22}$$

The initial quantum gas at zero-temperature is turned, at long times, into a low-temperature and low-density classical gas. Note that at long times $\mu\gg k_BT$, which is consistent with the proposed Maxwell-Boltzmann interpretation.

## 3.2 Long-time behaviour

We define the function $v(\tilde{t})=\int_0^{\tilde{t}}\tilde{n}(\tilde{t}')d\tilde{t}'$ and taking the integral on $\tilde{k}\in[-\pi,\pi]$ of Eq. (21) we obtain:

$$\int_{-\pi}^{\pi}n_{\tilde{k}}(\tilde{t})d\tilde{k}=2\pi\tilde{n}(\tilde{t})=\sqrt{\tilde{n}(\tilde{t})}\int_{-\pi}^{\pi}e^{-2\pi\tilde{k}^2v(\tilde{t})}d\tilde{k}; \tag{23}$$

in terms of the function $v(\tilde{t})$ we have:

$$\sqrt{\tilde{n}(\tilde{t})}=\sqrt{\frac{\partial}{\partial\tilde{t}}v(\tilde{t})}=\frac{1}{2\pi}\int_{-\pi}^{\pi}e^{-2\pi\tilde{k}^2v(\tilde{t})}d\tilde{k}=\frac{1}{2\pi}\sqrt{\frac{1}{2v(\tilde{t})}}\text{Erf}[\sqrt{2\pi v(\tilde{t})}\pi]. \tag{24}$$

A discussion of the properties of the system at short times is in Appendix B. At long time, we expect that $v(\tilde{t})\to\infty$ and consequently we insert in Eq. (24) the value $\lim_{x\to\infty}\text{Erf}[x]=1$. We obtain:

$$\frac{\partial}{\partial\tilde{t}}v(\tilde{t})=\frac{1}{8\pi^2}\frac{1}{v(\tilde{t})},\qquad\Rightarrow\qquad v(\tilde{t})=\sqrt{\frac{1}{4\pi^2}}\sqrt{\tilde{t}}. \tag{25}$$

The statement used in the previous section to discuss the variance of $n_{\tilde{k}}(\tilde{t})$ is thus proved. By differentiating with respect to $\tilde{t}$, we find the asymptotic behaviour of the density, which for clarity we present here also in dimensional units:

$$\tilde{n}(\tilde{t})=\frac{1}{4\pi}\sqrt{\frac{1}{\tilde{t}}};\qquad n(t)=n_{\text{in}}\times\frac{1}{4\pi}\sqrt{\frac{1}{n_{\text{in}}^3\Gamma_{\text{eff}}t}}. \tag{26}$$

We find a long-time behaviour characterised by $n(t)\sim t^{-1/2}$, similarly to what has already been found in the analogous lattice problem [36]. In order to appreciate the interest of this result, it is useful to compare it with the mean-field prediction $n(t)\sim t^{-1}$ obtained from the equation $\partial_t n=-\kappa n^2$, which follows from the approximation of uncorrelated bosons. Indeed, the correct equation that describes the population of the homogeneous gas under local two-body losses reads [34, 37]

$$\partial_t n(t)=-\kappa\,n(t)^2g^{(2)}(0),\qquad\text{with}\quad g^{(2)}(0)=\frac{\langle\hat{n}(\hat{n}-1)\rangle_t}{n(t)^2}. \tag{27}$$

In weakly-correlated bosonic gases, it is often the case that $g^{(2)}(0)=1$ is a good approximation (uncorrelated bosons), and the mean-field behaviour is recovered.

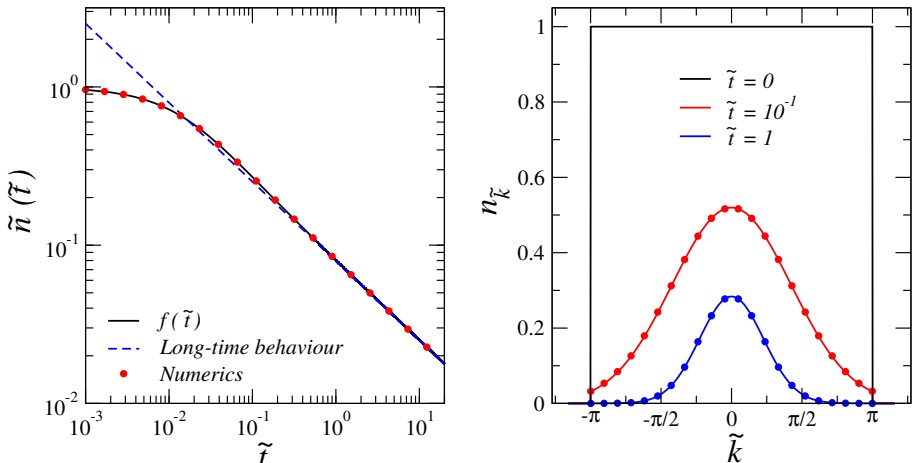

Figure 1: Left panel: Comparison of the numerically-computed normalised density $\tilde{n}(\tilde{t})$ (red dots) with the long-time behaviour given by Eq. (26) (blue dashed line). The black solid line represents Eq. (28), which faithfully reproduces the entire dynamics of $\tilde{n}(\tilde{t})$. Right panel: Density profile in $\tilde{k}$-space plotted for $\tilde{t} = 0$ (black), 0.1 (red), and 1 (blue). The circles represent the $n_{\tilde{k}}$ for $\tilde{t} = 0.1$ and $\tilde{t} = 1$ computed with the numerical integration of the Eqs. (18). The solid lines are obtained by taking $\tilde{n}(\tilde{t})$ from the numerical results and inserting it in Eq. (21).

The typical decay time scales as $(n_{in}^3 \Gamma_{eff})^{-1}$, so that a dense gas has a shorter lifetime than a dilute one. To get a more concrete idea about this time-scale, we consider once more the example of the $^{87}$Rb gas discussed above: for an initial density of $1 \times 10^7$ m$^{-1}$ we obtain a typical time of $2 \times 10^{-3}$ s. The Zeno effect can be fully appreciated if one compares this value with the original typical decay time of the gas: $(\gamma n_{in})^{-1} \sim 5 \times 10^{-6}$ s.

## 3.3 Numerical solution

In order to test the previous predictions, we have solved the Eqs. (18) with a 4$^{th}$-order Runge-Kutta numerical algorithm; the integration step is $d\tilde{t} = 10^{-3}$ and $N_{step} = 2 \cdot 10^4$, while we have discretised the $\tilde{k}$ space in $10^3$ points in the interval $[-\pi, \pi]$. Comparing with the results obtained employing $d\tilde{t} = 10^{-4}$, we estimate a relative error of $10^{-6}$, completely negligible for our purposes. The results are summarized in Fig. 1. We first compare the numerically-computed density $\tilde{n}(\tilde{t})$ with the long-time behaviour given by Eq. (26); the latter faithfully describes the behaviour of $\tilde{n}(\tilde{t})$ even for values of $\tilde{t}$ that are of order $10^{-1}$. Concerning the $n_{\tilde{k}}$, we observe that the initial Fermi-sea at $\tilde{t} = 0$ first evolves into a Gaussian that is truncated at $\tilde{k} = \pm\pi$; then, for sufficiently long times, it is possible to disregard this truncation and interpret the distribution as a Maxwell-Boltzmann classical distribution.

## 3.4 An analytical formula for $\tilde{n}(\tilde{t})$

We conclude this section proposing the following formula to describe the dynamics of $\tilde{n}(\tilde{t})$:

$$f(\tilde{t}) = \frac{\sqrt{1 + A\tilde{t}}}{1 + B\tilde{t}}, \tag{28}$$

where $A$ and $B$ are two coefficients to be determined. Notice that the same formula has been proposed in Ref. [36]. This functional form is motivated by the fact that it can capture both the short-time and the long-time behaviours. The short-time behaviour is discussed in Appendix B,

and in particular in Eq. (66) we show that $\tilde{n}(\tilde{t}) \sim 1 - \alpha \tilde{t}$. The long-time behaviour is discussed in Eq. (26), where $\tilde{n}(\tilde{t}) \sim (4\pi)^{-1} \tilde{t}^{-1/2}$. By comparing our ansatz with the two exact limits, we obtain two conditions for the coefficients $A$ and $B$ that are solved by the following values (details are in Appendix C):

$$A = -\frac{8}{3}\pi^2 \left[ 2\left(-6 + \sqrt{36-6\pi}\right) + \pi \right] \simeq 15.15\ldots, \qquad B = 4\pi\sqrt{A} \simeq 48.91\ldots. \tag{29}$$

In Fig. 1 left panel we compare the numerically-computed $\tilde{n}(\tilde{t})$ with Eq. (28); the plot shows that our formula describes very well the numerical data.

## 4  The harmonic confinement

We now include in our discussion the presence of an external potential, focusing on the experimentally-relevant case of a harmonic confinement:

$$V(x) = \frac{1}{2} m\omega^2 x^2. \tag{30}$$

In order to generalise the previous approach to an inhomogeneous situation, we employ the local-density approximation (LDA) and promote each fermionic momentum occupation number to a space-dependent quantity: $n_k(t) \to f(x,k,t)$, where $f(x,k,t)$ should be intended as an average over a small macroscopic region centered around $x$. The spatial density is obtained integrating over all possible momenta:

$$n(x) = \frac{1}{2\pi} \int_{-\infty}^{+\infty} f(x,k,t)\mathrm{d}k. \tag{31}$$

The time evolution of $f(x,k,t)$ is dictated by a Boltzmann-like equation that includes the effect of losses:

$$\frac{\partial f(x,k,t)}{\partial t} + \frac{\hbar k}{m} \frac{\partial f(x,k,t)}{\partial x} + \frac{F(x)}{\hbar} \frac{\partial f(x,k,t)}{\partial k} = -\Gamma_{\mathrm{eff}} \int_{-\infty}^{+\infty} (k-q)^2 f(x,k,t) f(x,q,t)\mathrm{d}q. \tag{32}$$

The force field depends on the potential; in the case of the harmonic potential: $F(x) = -\partial_x V(x) = -m\omega^2 x$.

The initial condition can be determined using the LDA for the equilibrium density profile $n_{\mathrm{in}}(x)$ of a gas in a harmonic potential:

$$n_{\mathrm{in}}(x) = \begin{cases} \frac{R}{\ell_{\mathrm{HO}}^2 \pi} \sqrt{1 - \frac{x^2}{R^2}} & \text{for } x \in [-R, R], \\ 0 & \text{otherwise;} \end{cases} \qquad R = \sqrt{2N_{\mathrm{in}}\ell_{\mathrm{HO}}^2}. \tag{33}$$

Here, $R$ represents the LDA radius of the gas, $N_{\mathrm{in}}$ is the initial number of particles and $\ell_{\mathrm{HO}} = \sqrt{\hbar/(m\omega)}$ is the harmonic oscillator length associated to the trap. The initial condition for the Boltzmann-like equation reads:

$$f(x,k,0) = \begin{cases} 1 & \text{for } x \in [-R, R] \text{ and } k \in [-\pi n_{\mathrm{in}}(x), \pi n_{\mathrm{in}}(x)]; \\ 0 & \text{otherwise.} \end{cases} \tag{34}$$

This approach is an example of the treatments based on the so-called generalised hydrodynamics that has been recently introduced to discuss the dynamics of one-dimensional integrable models with Bethe-ansatz techniques.

## 4.1 Dimensionless units

In order to get a better understanding of the problem, we introduce the following dimensionless variables:

$$\tilde{x} = \frac{1}{R}x; \qquad \tilde{k} = \frac{\ell_{\mathrm{HO}}^2}{R}k. \tag{35}$$

Notice that $\tilde{k}$ has a different definition with respect to the one presented in Sec. 3. This choice is motivated by the fact that in these units the initial condition (34) takes the particularly simple form of a circle with unitary radius:

$$f(\tilde{x}, \tilde{k}, 0) = \begin{cases} 1 & \text{for } \tilde{x}^2 + \tilde{k}^2 \le 1; \\ 0 & \text{otherwise.} \end{cases} \tag{36}$$

Also the Boltzmann-like equation (32) gets an intuitive form when the $\tilde{x}$ and $\tilde{k}$ coordinates are employed. We rescale time introducing:

$$\tilde{t} = \tilde{\Gamma}t, \qquad \text{with} \quad \tilde{\Gamma} = \Gamma_{\mathrm{eff}}\left(\frac{2m\omega N_{\mathrm{in}}}{\hbar}\right)^{\frac{3}{2}} = \Gamma_{\mathrm{eff}}\left(\frac{R}{\ell_{\mathrm{HO}}^2}\right)^3 = \Gamma_{\mathrm{eff}}\pi^3 n_{\mathrm{in}}(0)^3; \tag{37}$$

and we obtain:

$$\frac{\partial f(\tilde{x}, \tilde{k}, \tilde{t})}{\partial \tilde{t}} = -\frac{\omega}{\tilde{\Gamma}}\left(\tilde{k}\frac{\partial f(\tilde{x}, \tilde{k}, \tilde{t})}{\partial \tilde{x}} - \tilde{x}\frac{\partial f(\tilde{x}, \tilde{k}, \tilde{t})}{\partial \tilde{k}}\right) - \int_{-\infty}^{+\infty}(\tilde{k} - \tilde{q})^2 f(\tilde{x}, \tilde{k}, \tilde{t})f(\tilde{x}, \tilde{q}, \tilde{t})\mathrm{d}\tilde{q}. \tag{38}$$

As it is standard in the classical motion of the harmonic oscillator in phase space, in the absence of losses the initial unitary circle is remapped onto itself and rotates with a period $T_\omega = 2\pi/\omega$. Note that the losses do not scatter momenta outside the unitary circle, so that the dynamics remains confined within it. The ratio $\omega/\tilde{\Gamma}$ emerges as the relevant parameter that measures the competition between the harmonic confinement and two-body losses.

Similarly to what we have done in the homogeneous case, our focus will be the time evolution of the rescaled number of atoms $\tilde{N}$, defined as $\tilde{N} = N/N(0)$; since the area of the initial circle is $\pi$, we have:

$$\tilde{N}(\tilde{t}) = \frac{1}{\pi}\int_{-\infty}^{+\infty}\int_{-\infty}^{+\infty} f(\tilde{x}, \tilde{k}, \tilde{t})\,\mathrm{d}\tilde{x}\,\mathrm{d}\tilde{k}. \tag{39}$$

## 4.2 Numerical solution

We simulate Eq. (38) with a numerical algorithm; details about the actual implementation are reported in Appendix D. Our results for $f(\tilde{x}, \tilde{k}, \tilde{t})$ are presented in Fig. 2 for four different values of $\omega/\tilde{\Gamma}$. For some values of $\omega/\tilde{\Gamma}$ a link to the animated version of the dynamics is provided in footnote[1]. The time dependence of the integrated and normalised population $\tilde{N}(\tilde{t})$ is reported in Fig. 3; the observation of $\tilde{N}(\tilde{t})$ shows the existence of two limiting behaviours, for small and large values of $\omega/\tilde{\Gamma}$, dubbed *weak-confinement* and *strong-confinement* regimes, whose description will be the object of the next sections.

To make the discussion more concrete, we identify the value of the trapping frequency that separates the two regimes. The relation $\omega/\tilde{\Gamma} = 1$ yields the following equation for the trap frequency:

$$\omega = \frac{1}{\Gamma_{\mathrm{eff}}^2}\left(\frac{\hbar}{2mN_{\mathrm{in}}}\right)^3. \tag{40}$$

---

[1]To access the evolution of $f(\tilde{x}, \tilde{k}, \tilde{t})$ for $\omega/\tilde{\Gamma} = 0.2$ click **here** and for $\omega/\tilde{\Gamma} = 1$ click **here**.



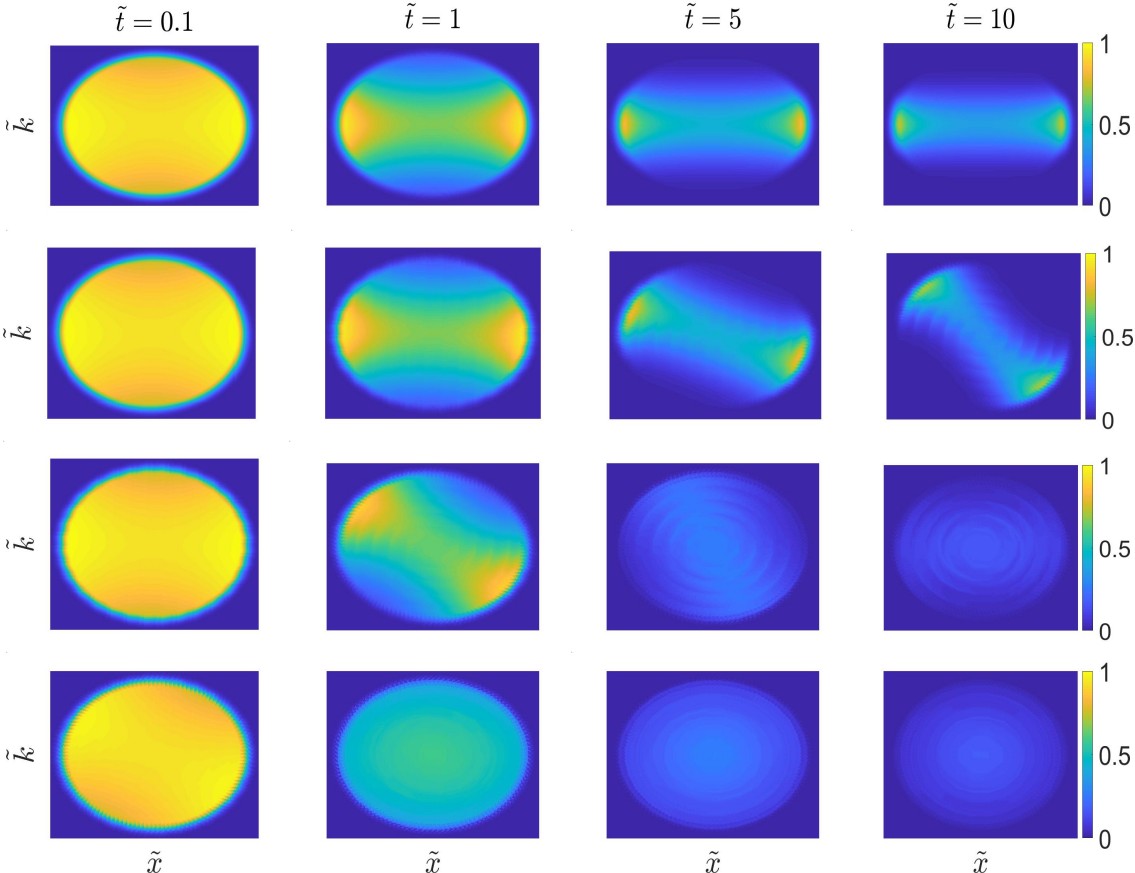

Figure 2: Snapshots of the density function $f(\tilde{x}, \tilde{k}, \tilde{t})$ in the $\tilde{x} - \tilde{k}$ phase space for different values of $\tilde{t} = 0.1$, 1, 5 and 10. Starting from the first row to the fourth one we set $\omega/\tilde{\Gamma} = 0$, $\omega/\tilde{\Gamma} = 0.1$, $\omega/\tilde{\Gamma} = 1$ and $\omega/\tilde{\Gamma} = 10$, respectively. The circle where $f(\tilde{x}, \tilde{k}, \tilde{t})$ is different from zero has unit radius (ticks not shown). Spirals are a numerical artefact due to the discretisation of the phase space, for further details see Appendix D.

Taking the numerical values obtained with the previous numerical estimates and an initial number of atoms $N_{\mathrm{in}} = 10^3$ we find $\omega = 0.22\,\mathrm{s}^{-1}$; this is an extremely small trapping frequency, given that those implemented in standard labs are of order $10^2\,\mathrm{s}^{-1}$. The observation of the weak confinement regime thus requires the use of a more dilute and less populated gas. If we consider the same parameters used in the previous estimate but take a lower density value, $n_{\mathrm{in}} = 5 \times 10^6\,\mathrm{m}^{-1}$, and an initial population of $N_{\mathrm{in}} = 100$ bosons, we obtain $\omega = 226.42\,\mathrm{s}^{-1}$.

### 4.3 The limit of weak confinement: $\omega/\tilde{\Gamma} \ll 1$

We begin with the analysis of the data in Fig. 2 for $\omega/\tilde{\Gamma} \le 1$, plotted in the first two lines. We observe that the gas depletion is stronger for higher values of $\tilde{k}$, a phenomenon that was characterising also the dynamics of the homogeneous gas, see Fig. 1. Moreover, in this inhomogeneous scenario, the losses are more effective in the centre of the trap, where the density is higher; a faster dynamics at higher densities was also observed in the homogeneous system, see Eq. (26). At the beginning of the dynamics the most long-lived population is thus located at the edges of the trap, where the density is lower and momenta are smaller. This is the first consequence of the presence of the trap: it changes the density profile of the gas with a faster

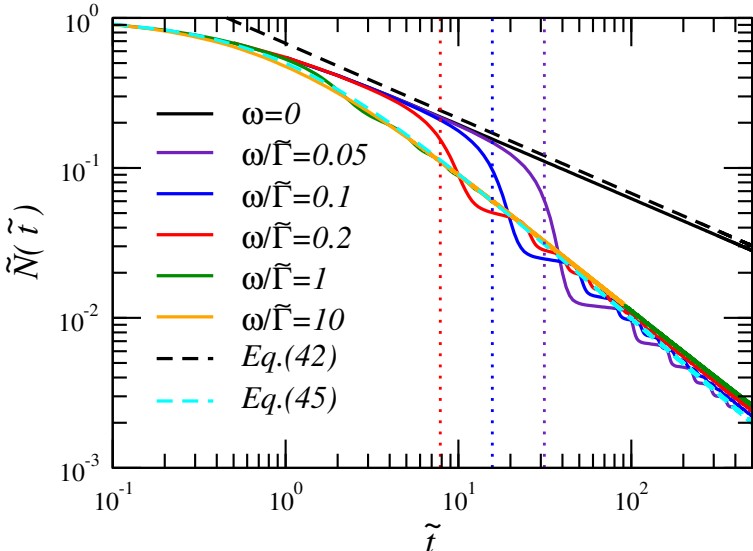

Figure 3: Time-evolution of the normalized density $\tilde{N}(\tilde{t})$ for different values of the ratio $\omega/\tilde{\Gamma}$. Black and cyan dashed lines are theoretical prediction for the weak and strong confinement case, Eq. (42) and Eq. (45) respectively. Vertical dashed lines mark the threshold time $t_{tr} = T_\omega/4$ for $\omega/\tilde{\Gamma} = 0.05, 0.1, 0.2$ (purple, blue and red line, respectively).

depletion of the population at the centre.

In order to present a quantitative theory of the short-time dynamics in the weak-confinement limit, we present an analytical solution for the case $\omega = 0$. Note that we are setting $\omega = 0$ in Eq. (38) but we are keeping the initial condition (36), which is shaped by the presence of the trap. In order to better visualize the depletion phenomenon in real space, in Fig. 4 we show the behavior of the spatial density

$$\tilde{N}(\tilde{x}, \tilde{t}) = \frac{1}{\pi} \int_{-\infty}^{+\infty} f(\tilde{x}, \tilde{k}, \tilde{t}) \, d\tilde{k}, \tag{41}$$

(left panel) and of its rescaled version $\tilde{N}(\tilde{x}, \tilde{t})/\tilde{N}(0, \tilde{t})$ (right panel), showing how the dissipative evolution leads to a spatial profile much denser at the boundaries with respect to the center of the trap.

Thanks to the LDA, we can model the gas as composed of several independent and homogeneous subparts located at different points of the trap and with different initial densities. According to our study of the homogeneous lossy gas in Sec. 3, each of these subparts features a long-time decay proportional to $\tilde{t}^{-\frac{1}{2}}$. The long-time behaviour of the trapped gas is obtained by integrating all these contributions; after some algebra reported in Appendix E we obtain:

$$\tilde{N}(\tilde{t}) \sim \frac{1}{2} \frac{\Gamma(3/4)}{\Gamma(5/4)} \frac{1}{\sqrt{\tilde{t}}}, \tag{42}$$

and $\Gamma(x)$ is the Euler $\Gamma$ function. This result is plotted in Fig. 3, where we observe that it reproduces quantitatively the long-time behaviour of the gas.

## 4.4 From weak to strong confinement

In the weak-confinement regime, once the inhomogeneous density profile shown in Fig. 4 has been created, the presence of the harmonic trap has an additional effect and induces a rotation

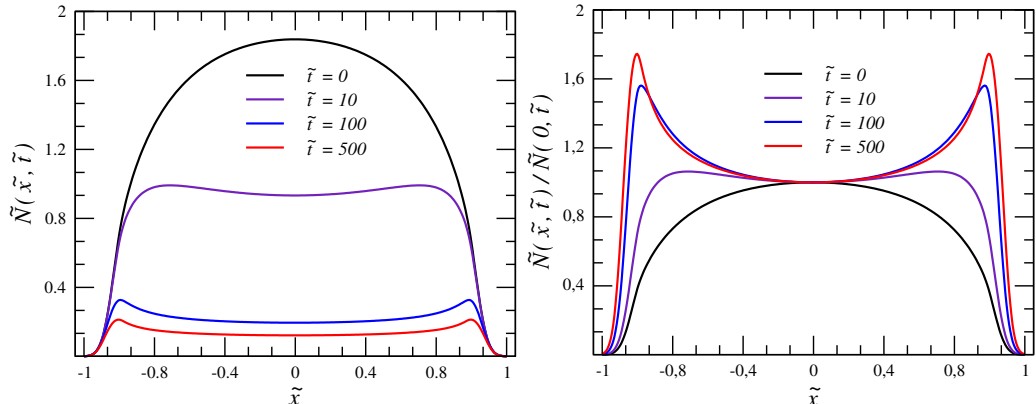

Figure 4: Spatial density profile $\tilde{N}(\tilde{x}, \tilde{t})$ (left panel) and its rescaled version $\tilde{N}(\tilde{x}, \tilde{t})/\tilde{N}(0, \tilde{t})$ for $\tilde{t} = 0, 10, 100$ and $500$ and $\omega = 0$.

in phase space (see for instance Fig. 2 for $\omega/\tilde{\Gamma} = 0.1$, second line). At time $t \sim T_\omega/4$ the long-lived bosons located initially at the edges of the trap have moved to the centre, where they are quickly lost because at the centre of the trap the loss mechanism is more effective (see for instance Fig. 2 for $\omega/\tilde{\Gamma} = 1.0$, third line). The time $t \sim T_\omega/4$ thus marks the onset of a novel behaviour that is clearly displayed in the plots of $\tilde{N}(\tilde{t})$ reported in Fig. 3. Since the time $T_\omega/4$ corresponds to the rescaled time $\tilde{t} = \pi\tilde{\Gamma}/(2\omega)$, we observe that when the initial confinement is weak, $\omega/\tilde{\Gamma} \lesssim 1$, $\tilde{N}(\tilde{t})$ departs from the $\omega = 0$ curve and after some oscillations collapses onto a new curve that describes the depopulation in the strong-confinement regime.

We can make sense of this transition by comparing the frequency of the rotation in phase space with the instantaneous decay-time of the gas. We introduce the following adimensional parameter:

$$r(t) = \frac{1}{\omega} \times \left| \frac{1}{\tilde{N}(\tilde{t})} \frac{d\tilde{N}(\tilde{t})}{dt} \right|. \tag{43}$$

For $r \gg 1$ we have a weakly-confined gas, whereas for $r \ll 1$ we have a strongly-confined gas. For a gas that is initially weakly-confined, using the formula in Eq. (42) we obtain that $r(t) = 1/(2\omega t)$, a decreasing function of time. Thus, there must be a time $t_{tr}$ at which the typical decay-time becomes longer than the trapping frequency. For $t \gtrsim t_{tr}$ the gas becomes effectively strongly-confined and the previous approximation cannot hold anymore: the gas cannot be described by Eq. (42). If we set this characteristic time as a quarter of the harmonic period, i.e. $\omega t_{tr} = \pi/2$, we find

$$t_{tr} = \frac{T_\omega}{4}, \tag{44}$$

which gives $r(t_{tr}) = \pi$, i.e. a value of order unity for the parameter $r$ at $t = t_{tr}$. Thus, with completely different arguments, we have found again the importance of the harmonic period in signaling the transition from a weakly-confined behaviour to a strongly-confined one.

## 4.5 The limit of strong confinement: $\omega/\tilde{\Gamma} \gg 1$

We observe in Fig. 3 that for $\omega/\tilde{\Gamma} \geq 1$ all curves collapse onto a universal function. In this limit, the dynamics is strongly determined by the trap, and it results in a behaviour that is well described by the formula:

$$\tilde{N}(\tilde{t}) = \frac{1}{1 + \tilde{t}}, \tag{45}$$

as it is shown in the plot.

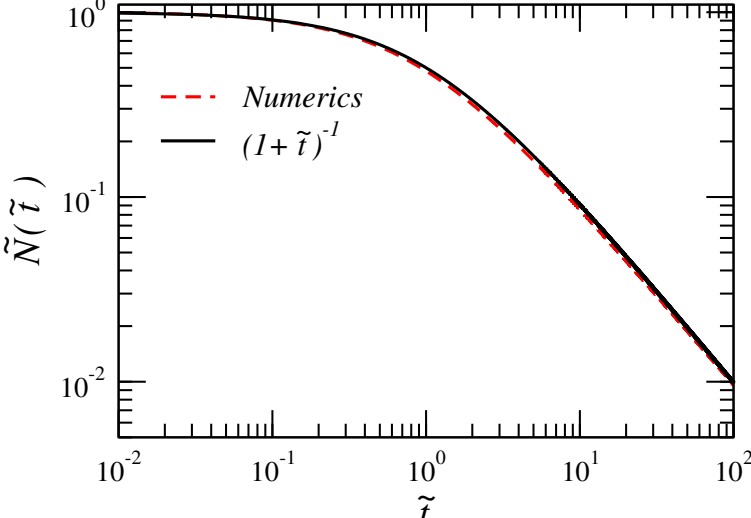

Figure 5: Numerical solution of Eq. (46) for $\tilde{N}(\tilde{t})$ (red-dashed line) compared with the form $\tilde{N}(\tilde{t}) = (1 + \tilde{t})^{-1}$ (solid black line).

We now present a simple model that can explain the formula in Eq. (45). For $\omega/\tilde{\Gamma} \geq 1$, the inhomogeneities in the momentum distribution function that are created by the losses are rapidly washed out by the action of the trap, that rotates the $f(\tilde{x}, \tilde{k}, \tilde{t})$ in phase space (see for instance Fig. 2 for $\omega/\tilde{\Gamma} = 10$, fourth line). Since this rotation takes place on the shortest time-scale of the problem, we introduce a novel distribution function $g(\tilde{e}, \tilde{t})$ that does not depend on $\tilde{x}$ and $\tilde{k}$ in a separate way, but only on the dimensionless energy $\tilde{e} = \tilde{x}^2 + \tilde{k}^2$ that is conserved by the harmonic-oscillator dynamics; normalisation is ensured by the following relation: $\tilde{N}(\tilde{t}) = \frac{1}{\pi} \int_0^\infty g(\tilde{e}, \tilde{t}) d\tilde{e}$. This "radial" symmetry in phase space is well highlighted by the plots in Fig. 2 for $\omega/\tilde{\Gamma} = 10$. In the next section, we show how to derive from Eqs. (38) the rate equation obeyed by $g(\tilde{e}, \tilde{t})$, which we anticipate here:

$$\frac{\partial g(\tilde{e}, \tilde{t})}{\partial \tilde{t}} = -\frac{1}{\pi^2} \int_0^1 d\tilde{\varepsilon}\, g(\tilde{e}, \tilde{t}) g(\tilde{\varepsilon}, \tilde{t}) \int_{-\min(\sqrt{\tilde{e}}, \sqrt{\tilde{\varepsilon}})}^{+\min(\sqrt{\tilde{e}}, \sqrt{\tilde{\varepsilon}})} d\tilde{x} \left( \sqrt{\frac{\tilde{\varepsilon} - \tilde{x}^2}{\tilde{e} - \tilde{x}^2}} + \sqrt{\frac{\tilde{e} - \tilde{x}^2}{\tilde{\varepsilon} - \tilde{x}^2}} \right). \tag{46}$$

We did not find an analytical solution of Eq. (46), but we could easily produce a numerical solution for the initial condition:

$$g(e, 0) = 2\pi e, \tag{47}$$

which satisfies $\int_0^1 g(e, 0) de = \pi$, the area of the circle. We display in Fig. 5 the numerically-computed $\tilde{N}(\tilde{t})$; the result is well described by Eq. (45). The appearance of this novel behaviour leads to an interesting qualitative observation: the presence of a harmonic confinement *accelerates* the depletion of the gas. Indeed, in the presence of a trap the depletion is eventually scaling as $1/t$, whereas for a homogeneous system it scales as $1/t^{1/2}$. Clearly, the previous is a statement concerning the long-time asymptotics: the possibility of observing experimentally the decay as $1/t^{1/2}$ is not ruled out if one works with a shallow trap and in the appropriate time regime.

Finally, a last consistency check. If we compute the parameter $r(t)$ for the gas in the strong-confinement limit using the formula in Eq. (45), we obtain $r(t) = 1/(\omega t)$. Again, we encounter a decreasing function of time. Thus, if we start with $r \ll 1$, the system is expected to remain in the strongly-confined regime, and the description is self-consistent at all considered times.

## 4.6 Rate equations in the strong-confinement limit $\omega/\tilde{\Gamma} \gg 1$

As discussed above, in the strong-confinement limit we consider the phase-space density function $g(\tilde{e}, \tilde{t})$, and the normalised population reads $\tilde{N}(\tilde{t}) = \frac{1}{\pi} \int_0^1 g(\tilde{e}, \tilde{t}) d\tilde{e}$. We now identify the evolution equation that is satisfied by $g(\tilde{e}, \tilde{t})$.

First, we observe that the part of the Boltzmann-like equation that is responsible for the classical motion in phase space leaves $g(\tilde{e}, \tilde{t})$ unchanged. Indeed:

$$\tilde{k} \frac{\partial g(\tilde{e}, \tilde{t})}{\partial \tilde{x}} - \tilde{x} \frac{\partial g(\tilde{e}, \tilde{t})}{\partial \tilde{k}} = \tilde{k} \frac{\partial \tilde{e}}{\partial \tilde{x}} \frac{\partial g}{\partial \tilde{e}} - \tilde{x} \frac{\partial \tilde{e}}{\partial \tilde{k}} \frac{\partial g}{\partial \tilde{e}} = 0. \tag{48}$$

We then move to the loss term, which is local in space. Since the particles with energy $\tilde{\epsilon}$ are assumed to be uniformly spread in the region of the phase space with energy $\tilde{\epsilon}$, we need to introduce the probability density $p(\tilde{x}, \tilde{\epsilon})$ that describes the fraction of particles with energy $\tilde{\epsilon}$ that are located at position $\tilde{x}$. With simple calculations we obtain:

$$p(\tilde{x}, \tilde{\epsilon}) = \frac{1}{\pi} \frac{1}{\sqrt{\tilde{\epsilon} - \tilde{x}^2}}. \tag{49}$$

It is a good probability density, and indeed is satisfies $\int_{-\sqrt{\tilde{\epsilon}}}^{\sqrt{\tilde{\epsilon}}} p(\tilde{x}, \tilde{\epsilon}) d\tilde{x} = 1$, where we have used that the extrema of the classically-allowed region are $\pm\sqrt{\tilde{\epsilon}}$. With this notation, the number of particles with energy in $[\tilde{e}, \tilde{e} + d\tilde{e}]$ located in the infinitesimal space region $[\tilde{x}, \tilde{x} + d\tilde{x}]$ is:

$$g(\tilde{e}, \tilde{t}) p(\tilde{e}, \tilde{x}) d\tilde{e} d\tilde{x}. \tag{50}$$

Let us now focus on the loss of particles with energy $\tilde{e}$ due to the presence of particles with energy $\tilde{\epsilon}$. Recall that losses depend on the squared momentum difference $(\tilde{k} - \tilde{q})^2$ at the same position $\tilde{x}$. The particles with energy $\tilde{e}$ at position $\tilde{x}$ can have two equally-likely momenta: $\pm\tilde{k}_{\tilde{e}, \tilde{x}}$, with $\tilde{k}_{\tilde{e}, \tilde{x}} = +\sqrt{\tilde{e} - \tilde{x}^2}$. Note that $\pi \times \tilde{k}_{\tilde{e}, \tilde{x}} \times p(\tilde{e}, \tilde{x}) = 1$. The same is true for particles with energy $\tilde{\epsilon}$ at position $\tilde{x}$. Thus, concerning the loss-rate momentum dependence $(\tilde{k} - \tilde{q})^2$, four different combination of momenta are then possible:

$$\begin{cases} +\tilde{k}_{\tilde{e}, \tilde{x}} & +\tilde{k}_{\tilde{\epsilon}, \tilde{x}} & \longrightarrow & (\tilde{k}_{\tilde{e}, \tilde{x}} - \tilde{k}_{\tilde{\epsilon}, \tilde{x}})^2 \\ +\tilde{k}_{\tilde{e}, \tilde{x}} & -\tilde{k}_{\tilde{\epsilon}, \tilde{x}} & \longrightarrow & (\tilde{k}_{\tilde{e}, \tilde{x}} + \tilde{k}_{\tilde{\epsilon}, \tilde{x}})^2 \\ -\tilde{k}_{\tilde{e}, \tilde{x}} & +\tilde{k}_{\tilde{\epsilon}, \tilde{x}} & \longrightarrow & (\tilde{k}_{\tilde{e}, \tilde{x}} + \tilde{k}_{\tilde{\epsilon}, \tilde{x}})^2 \\ -\tilde{k}_{\tilde{e}, \tilde{x}} & -\tilde{k}_{\tilde{\epsilon}, \tilde{x}} & \longrightarrow & (\tilde{k}_{\tilde{e}, \tilde{x}} - \tilde{k}_{\tilde{\epsilon}, \tilde{x}})^2 \end{cases} \tag{51}$$

each with probability density

$$\frac{g(\tilde{e}, \tilde{t}) p(\tilde{x}, \tilde{t})}{2} \times \frac{g(\tilde{\epsilon}, \tilde{t}) p(\tilde{x}, \tilde{t})}{2}. \tag{52}$$

Let us first consider $\tilde{\epsilon} < \tilde{e}$. Here, we have to restrict the integration over $\tilde{x}$ to $\pm\sqrt{\tilde{\epsilon}}$, which is the maximal spatial extension of the particles with smallest energy; for $|\tilde{x}| > \sqrt{\tilde{\epsilon}}$ no interaction is possible. The loss of particles with energy $\tilde{e}$ due to the presence of particles with energy $\tilde{\epsilon}$ reads:

$$-\int_{-\sqrt{\tilde{\epsilon}}}^{\sqrt{\tilde{\epsilon}}} g(\tilde{e}, \tilde{t}) g(\tilde{\epsilon}, \tilde{t}) \frac{p(\tilde{x}, \tilde{e})}{2} \frac{p(\tilde{x}, \tilde{\epsilon})}{2} \left[ 2(\tilde{k}_{\tilde{e}, \tilde{x}} - \tilde{k}_{\tilde{\epsilon}, \tilde{x}})^2 + 2(\tilde{k}_{\tilde{e}, \tilde{x}} + \tilde{k}_{\tilde{\epsilon}, \tilde{x}})^2 \right] d\tilde{x}. \tag{53}$$

This expression can be simplified, because $(\tilde{k}_{\tilde{e}, \tilde{x}} - \tilde{k}_{\tilde{\epsilon}, \tilde{x}})^2 + (\tilde{k}_{\tilde{e}, \tilde{x}} + \tilde{k}_{\tilde{\epsilon}, \tilde{x}})^2 = 2\tilde{k}_{\tilde{e}, \tilde{x}}^2 + 2\tilde{k}_{\tilde{\epsilon}, \tilde{x}}^2$. The generalisation to the situation $\tilde{\epsilon} > \tilde{e}$ is simple.

By integrating over $\tilde{\epsilon}$, we write:

$$\frac{\partial}{\partial \tilde{t}} g(\tilde{e}, \tilde{t}) = - \int_0^{\tilde{e}} \mathrm{d}\tilde{\epsilon}\, g(\tilde{e}, \tilde{t}) g(\tilde{\epsilon}, \tilde{t}) \int_{-\sqrt{\tilde{\epsilon}}}^{\sqrt{\tilde{\epsilon}}} p(\tilde{x}, \tilde{e}) p(\tilde{x}, \tilde{e})(\tilde{k}_{\tilde{e}, \tilde{x}}^2 + \tilde{k}_{\tilde{\epsilon}, \tilde{x}}^2)\mathrm{d}\tilde{x} +$$
$$- \int_{\tilde{e}}^1 \mathrm{d}\epsilon\, g(\tilde{e}, \tilde{t}) g(\tilde{\epsilon}, \tilde{t})\tilde{\epsilon} \int_{-\sqrt{\tilde{e}}}^{\sqrt{\tilde{e}}} p(\tilde{x}, \tilde{e}) p(\tilde{x}, \tilde{e})(\tilde{k}_{\tilde{e}, \tilde{x}}^2 + \tilde{k}_{\tilde{\epsilon}, \tilde{x}}^2)\mathrm{d}\tilde{x}. \tag{54}$$

With some algebra, we obtain

$$\frac{\partial}{\partial \tilde{t}} g(\tilde{e}, \tilde{t}) = -\frac{1}{\pi^2} \int_0^{\tilde{e}} \mathrm{d}\tilde{\epsilon}\, g(\tilde{e}, \tilde{t}) g(\tilde{\epsilon}, \tilde{t}) \int_{-\sqrt{\tilde{\epsilon}}}^{\sqrt{\tilde{\epsilon}}} \left( \sqrt{\frac{\tilde{\epsilon} - \tilde{x}^2}{\tilde{e} - \tilde{x}^2}} + \sqrt{\frac{\tilde{e} - \tilde{x}^2}{\tilde{\epsilon} - \tilde{x}^2}} \right) \mathrm{d}\tilde{x} +$$
$$-\frac{1}{\pi^2} \int_{\tilde{e}}^1 \mathrm{d}\tilde{\epsilon}\, g(\tilde{e}, \tilde{t}) g(\tilde{\epsilon}, \tilde{t}) \int_{-\sqrt{\tilde{e}}}^{\sqrt{\tilde{e}}} \left( \sqrt{\frac{\tilde{\epsilon} - \tilde{x}^2}{\tilde{e} - \tilde{x}^2}} + \sqrt{\frac{\tilde{e} - \tilde{x}^2}{\tilde{\epsilon} - \tilde{x}^2}} \right) \mathrm{d}\tilde{x}. \tag{55}$$

This equation is amenable to the compact writing which was presented in Eq. (46).

## 5 Conclusions and perspectives

In this article we have studied the effect of a harmonic confinement on the dynamics of a one-dimensional Bose gas subject to strong two-body losses. Thanks to the fermionisation that takes place in the regime of strong dissipation, the so-called quantum Zeno regime, we have developed a set of rate equations that describe the depopulation of each fermionic mode. In the homogeneous case we have predicted both the dynamics of the density of the gas, displaying an anomalous $t^{-\frac{1}{2}}$ decay, and the time-dependence of the fermionic momentum distribution function. This latter quantity, which is often called *distribution of rapidities*, is a measurable quantity, via an expansion in a one-dimensional setting. The main result of the article is that the presence of a harmonic confinement leads to a drastic modification of the dynamics of the gas; in particular, when the trap is strong, the density of the gas decreases as $t^{-1}$. We have shown the existence of a crossover time that signals the passage from the behaviour of a homogeneous-gas at short times to that of a strongly-confined gas at long times. Several of our predictions can be tested in state-of-the-art experiments.

Two directions of further study can be devised. The first one concerns the fact that during the dynamics the gas might leave the purely one-dimensional situation, and start to populate excited levels of the transverse confinement. The fact that we are working in the strongly-dissipative Zeno regime makes this not implausible because experiments are typically longer than standard ones. Usually, including higher energy bands in a numerical or analytical description can be extremely challenging, as it requires the inclusion of a novel degree of freedom, which enlarges exponentially the Hilbert space to be considered. In our case, instead, the rate-equation model scales linearly with the number of fermionic modes. As such, the inclusion of a higher-energy band within this framework is feasible and can be an important step to quantitatively describe the experimental data.

A second investigation direction concerns the study of strong and finite elastic interactions, which could be responsible for the scattering of the fermionic momenta in phase space. These processes could motivate the introduction of novel terms in the dynamics, so far neglected because they are less important than those discussed in the article. Among the processes that could be anticipated, we mention diffusive terms, originally introduced in Ref. [50], that have been shown to be eventually responsible for the thermalisation of the one-dimensional Bose gas in the presence of an external confinement [51]. The rate equations completely disregard

interactions between the fermionic momenta, which could be responsible for some forms of redistribution, and eventually change the decay of the gas. The physical importance of these terms motivates further studies in this direction.

## Acknowledgements

We gratefully acknowledge discussions with J. Beugnon, I. Bouchoule, J. Dubail, A. De Luca, J. De Nardis, F. Gerbier and D. Rossini. The discussion in Sec. 4.4 is based on a suggestion of the anonymous Reviewer 2 of SciPost, whom we warmly thank.

**Funding information**   We gratefully acknowledge funding from LabEx PALM (ANR-10-LABX-0039-PALM).

## A   Homogeneous Bose gas: derivations

We now discuss the derivation of Eqs. (19) and of Eq. (21) presented in Sec. 3. We consider the time evolution of $\tilde{n}(\tilde{t})$ written as:

$$\frac{\partial}{\partial \tilde{t}} \tilde{n}(\tilde{t}) = -\frac{1}{2\pi} \int_{-\pi}^{\pi} d\tilde{k} \frac{\partial}{\partial \tilde{t}} n_{\tilde{k}}(\tilde{t}). \tag{56}$$

We now insert the rate equations (18) obtaining:

$$\frac{\partial}{\partial \tilde{t}} \tilde{n}(\tilde{t}) = -\frac{1}{2\pi} \int_{-\pi}^{\pi} d\tilde{k} \int_{-\pi}^{\pi} d\tilde{q} \left(\tilde{k}^2 + \tilde{q}^2\right) n_{\tilde{k}}(\tilde{t}) n_{\tilde{q}}(\tilde{t}) = \tag{57}$$

$$= -2 \left( \frac{1}{2\pi} \int_{-\pi}^{\pi} d\tilde{k} \, n_{\tilde{k}}(\tilde{t}) \int_{-\pi}^{\pi} d\tilde{q} \, \tilde{q}^2 n_{\tilde{q}}(\tilde{t}) \right) = \tag{58}$$

$$= -2\tilde{n}(\tilde{t}) \int_{-\pi}^{\pi} d\tilde{q} \, \tilde{q}^2 n_{\tilde{q}}(\tilde{t}). \tag{59}$$

For what concerns the time evolution of $n_{\tilde{k}}(\tilde{t})$, the rate equations (18) can be recast into the following form:

$$\frac{\partial}{\partial \tilde{t}} n_{\tilde{k}}(\tilde{t}) = -2\pi \tilde{k}^2 n_{\tilde{k}}(\tilde{t}) \tilde{n}(\tilde{t}) - n_{\tilde{k}}(\tilde{t}) \int_{-\pi}^{\pi} d\tilde{q} \, \tilde{q}^2 n_{\tilde{q}}(\tilde{t}) = \tag{60}$$

$$= -2\pi \tilde{k}^2 n_{\tilde{k}}(\tilde{t}) \tilde{n}(\tilde{t}) + \frac{n_{\tilde{k}}(\tilde{t})}{2\tilde{n}(\tilde{t})} \partial_{\tilde{t}} \tilde{n}(\tilde{t}), \tag{61}$$

where in the last passage we used the previous result for $\partial_{\tilde{t}} \tilde{n}(\tilde{t})$. This concludes the derivation of Eqs. (19).

We want now to show that Eq. (20) is solved by Eq. (21). By integrating Eq. (20) one obtains:

$$\ln \left( \frac{n_{\tilde{k}}(\tilde{t})}{n_{\tilde{k}}(0)} \right) = \ln \sqrt{\left( \frac{\tilde{n}(\tilde{t})}{\tilde{n}(0)} \right)} - 2\pi \tilde{k}^2 \int_0^{\tilde{t}} \tilde{n}(\tilde{t}') d\tilde{t}'. \tag{62}$$

By means of logarithm properties and subsequent exponentiation we get:

$$n_{\tilde{k}}(\tilde{t}) = \sqrt{\tilde{n}(\tilde{t})} e^{-2\pi \tilde{k}^2 \int_0^{\tilde{t}} \tilde{n}(\tilde{t}') d\tilde{t}'}, \tag{63}$$

this consludes the derivation of Eq. (21).

## B Homogeneous Bose gas: short-time behaviour

We now characterise the behaviour of the gas at short time using Eq. (24), which is exact. At short time $v(\tilde{t}) \ll 1$ and we can use the series expansion for the $\text{Erf}(x) = 2x/\sqrt{\pi} - 2x^3/(3\sqrt{\pi})$; since $\tilde{n}(\tilde{t}) = \partial_{\tilde{t}} v(\tilde{t})$:

$$\frac{\partial}{\partial \tilde{t}} v(\tilde{t}) \approx 1 - \frac{4\pi^3}{3} v(\tilde{t}) + \dots \tag{64}$$

The solution of this differential equation at short times is:

$$v(\tilde{t}) = \tilde{t} - \frac{2\pi^3}{3} \tilde{t}^2 + \dots. \tag{65}$$

By differentiating with respect to $\tilde{t}$ we obtain the short-time behaviour of the population:

$$\tilde{n}(\tilde{t}) = \frac{\partial}{\partial \tilde{t}} v(\tilde{t}) = 1 - \frac{4\pi^3}{3} \tilde{t} + \dots \tag{66}$$

## C Homogeneous Bose gas: Analytical expressions for $n(\tilde{t})$ and $v(\tilde{t})$

We now discuss the derivation of the values of the coefficients $A$ and $B$ in Eq. (29). We consider both the short- and long-time behaviour of Eq. (28):

$$f(\tilde{t}) \underset{\tilde{t} \ll 1}{\simeq} 1 + \tilde{t}\left(\frac{A}{2} - B\right) + \mathcal{O}(\tilde{t}^2) \tag{67a}$$

$$f(\tilde{t}) \underset{\tilde{t} \gg 1}{\simeq} \frac{\sqrt{A}}{B}\sqrt{\frac{1}{\tilde{t}}}. \tag{67b}$$

By imposing that $f(\tilde{t})$ has the same short- and long-time behaviour of $n(\tilde{t})$, we obtain the following two conditions for $A$ and $B$:

$$\frac{A}{2} - B = \frac{4}{3}\pi^3; \qquad \frac{\sqrt{A}}{B} = \frac{1}{4\pi}. \tag{68}$$

The values in Eq. (29) solve this system of equations; the other solution of the system produce a function $f(\tilde{t})$ that does not reproduce the numerical data.

We can also present an analytical formula also for $v(\tilde{t})$:

$$v(\tilde{t}) = \frac{2}{B}\left(-1 + \sqrt{1 + A\tilde{t}}\right) + \frac{2\sqrt{B-A}}{B^{\frac{3}{2}}}\left(\text{arctanh}\sqrt{\frac{B}{B-A}} - \text{arctanh}\sqrt{\frac{B(1+A\tilde{t})}{B-A}}\right). \tag{69}$$

This expression is interesting because we can use it to see whether it satisfies the differential equation (24), and thus to check whether the analytical formula that we found is exact or not. We find that this is not the case.

## D The numerical implementation of the Boltzmann equation

In this appendix we present some details related to the numerical simulation of the dynamics induced by Eq. (38). We integrate it exploiting the second-order Suzuki-Trotter expansion of the differential operator governing the dynamics, namely the r.h.s. of Eq. (38):

$$f(\tilde{x}, \tilde{k}, \tilde{t} + d\tilde{t}) = \mathcal{D}^{\text{HO}}_{d\tilde{t}/2} \circ \mathcal{D}^{\text{loss}}_{d\tilde{t}} \circ \mathcal{D}^{\text{HO}}_{d\tilde{t}/2}\left[f(\tilde{x}, \tilde{k}, \tilde{t})\right] + \mathcal{O}(d\tilde{t}^2). \tag{70}$$

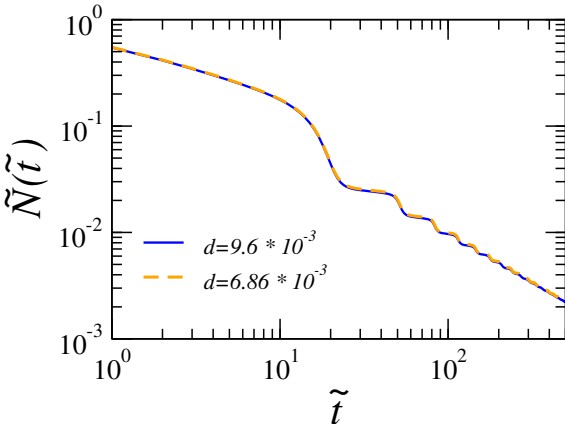

Figure 6: Comparison of the time-evolution of the normalized density $\tilde{N}(\tilde{t})$ for $\omega/\tilde{\gamma} = 0.1$ obtained for two different discretization of the phase space $d$, as indicated in the legend.

Here, $\mathcal{D}^{\text{HO}}_{d\tilde{t}/2}$ describes the dynamics induced by the trap, i.e. first term of the r.h.s. of Eq. (38), and $\mathcal{D}^{\text{loss}}_{d\tilde{t}/2}$ is responsible for the two-body losses, i.e. second term of the r.h.s. of Eq. (38). The action of the two operators $\mathcal{D}^{\text{HO}}_{d\tilde{t}/2}$ on the density distribution can be implemented *exactly* since it is a rigid rotation of the density function in the phase space $\tilde{x} - \tilde{k}$ of and angle $(\omega/\tilde{\gamma})d\tilde{t} = \omega dt$. The action of the the operator $\mathcal{D}^{\text{loss}}_{d\tilde{t}}$ is implemented with a fourth-order Runge-Kutta method with variable time-step.

We smoothened the initial condition in order to avoid numerical problems connected to the sharpness of the initial distribution. In particular we used:

$$f(\tilde{x}, \tilde{k}, 0) = \begin{cases} 1 & \tilde{x}^2 + \tilde{k}^2 \le 1 - \xi; \\ e^{-[\tilde{x}^2 + \tilde{k}^2 - (1-\xi)]^2/\xi} & \text{otherwise}, \end{cases} \tag{71}$$

with $\xi = 5d$. This smoothening is visible in the plots of Fig. 2.

For all the data shown in Fig. 2 we discretized the phase space with a grid of $250 \times 250$ pixels for the phase space $\tilde{x}, \tilde{k} \in [-1.2, 1.2]$. This discretization implies a resolution $d = d\tilde{x} = d\tilde{k} = 9.6 \cdot 10^{-3}$. We check that the results obtained for the time-evolution of the normalized density $\tilde{N}(\tilde{t})$ do not depend on the size of the pixels. In Fig.6 we show $\tilde{N}(\tilde{t})$ for two different values of $d$. The two curves collapse for the time window under consideration.

## E   Analytical solution for $\omega/\tilde{\Gamma} = 0$

We solve the Eq. (38) for $\omega/\tilde{\Gamma} = 0$. Whereas the presence of the trap is taken into account by the initial conditions, it is completely disregarded in the dynamics, so that it describes the situation where losses are extremely more important. The Boltzmann equation factorizes the dynamics at different $\tilde{x}$:

$$\frac{\partial f(\tilde{x}, \tilde{k}, \tilde{t})}{\partial \tilde{t}} = -\int_{-\sqrt{1-\tilde{x}^2}}^{+\sqrt{1+\tilde{x}^2}} (\tilde{k} - \tilde{q})^2 f(\tilde{x}, \tilde{k}, \tilde{t}) f(\tilde{x}, \tilde{q}, \tilde{t}) \mathrm{d}\tilde{q}, \tag{72}$$

where we have explicitly written that $f(x, q, t)$ is different from zero only for $x^2 + q^2 \leq 1$. By rescaling momenta and time as follows:

$$K = \frac{\pi}{\sqrt{1 - \tilde{x}^2}} \tilde{k}, \qquad Q = \frac{\pi}{\sqrt{1 - \tilde{x}^2}} \tilde{q}, \qquad T = \frac{(1 - \tilde{x}^2)^{\frac{3}{2}}}{\pi^3} \tilde{t}; \tag{73}$$

we can recast the Boltzmann-like equation in a form that we have already encountered, see Eq. (18) :

$$\frac{\partial}{\partial T} f(\tilde{x}, K, T) = -\int_{\pi}^{\pi} (K^2 + Q^2) f(\tilde{x}, K, T) f(\tilde{x}, Q, T) \mathrm{d}Q. \tag{74}$$

Using the results in Sec. 3, we obtain the long-time limit:

$$\frac{1}{2\pi} \int_{-\pi}^{\pi} f(\tilde{x}, K, T) \mathrm{d}K \sim \frac{1}{4\pi} \sqrt{\frac{1}{T}} \qquad \Rightarrow \qquad \int_{-\sqrt{1-x^2}}^{\sqrt{1-\tilde{x}^2}} f(\tilde{x}, \tilde{k}, \tilde{t}) \mathrm{d}\tilde{k} \sim \frac{\sqrt{\pi}}{2} \frac{1}{(1 - \tilde{x}^2)^{\frac{1}{4}}} \sqrt{\frac{1}{\tilde{t}}}. \tag{75}$$

We finally obtain the result reported in the main text:

$$\tilde{N}(\tilde{t}) = \frac{1}{\pi} \int_{-1}^{1} \int_{-\sqrt{1-\tilde{x}^2}}^{\sqrt{1-\tilde{x}^2}} f(\tilde{x}, \tilde{k}, \tilde{t}) \mathrm{d}\tilde{x} \mathrm{d}\tilde{k} = \frac{1}{\sqrt{4\pi\tilde{t}}} \int_{-1}^{1} \frac{1}{(1 - \tilde{x}^2)^{\frac{1}{4}}} \mathrm{d}\tilde{x} = \frac{1}{2} \frac{\Gamma(3/4)}{\Gamma(5/4)} \frac{1}{\sqrt{\tilde{t}}}. \tag{76}$$

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
