# Peer review of "The one-dimensional Bose gas with strong two-body losses: the effect of the harmonic confinement"

_SciPost Physics, doi:SciPost Phys. 12, 044 (2022)_

## Round 1 · Referee Report · Anonymous · 2021-7-30

Strengths
1 - Introduces and sets the research in its appropriate context, with a thorough explanation of where it sits in comparison to existing literature that are close in scope to this work.
2 - The results provide useful and experimentally relevant insights into the nature of strong two-body losses in homogenous and trapped 1D Bose gases.
3 - The discovery of the novel decay regime and effect of a confining potential on the two-body loss dynamics appear significant and original.
Weaknesses
1 - The presentation of some derivations (along with typographical errors) could be improved for ease of understanding and flow.
2- Overall the grammar and flow of concepts/content should be enhanced to increase the quality of communication of the research performed.
Report
The authors study the population dynamics of both a homogenous and harmonically confined 1D Bose gas with strong two-body losses using a set of rate equations applicable in the quantum Zeno regime (strong dissipation). In the homogenous case the results extend those derived for a gas on a lattice in Ref [36], as well as those for the continuum situation in Ref [37], and deviate from a naive mean-field prediction in the long-time limit. Their methodology allows them to extend their results to include a trapping potential and show how the gas is depleted in the experimentally relevant scenario of harmonic confinement. In this case, they predict that the depletion is accelerated and that a novel decay regime arrises where the behaviour of the gas is consistent with strong-confinement at long times, even when the gas is only weakly trapped.
Overall the reviewer finds the work worthy to be published, but only after revision of the manuscript to enhance the quality of communication of the results reported in the paper.
Some comments and questions for consideration:
1 - In the introduction, generalised hydrodynamics (GHD) is identified as a theory which successfully describes out-of-equilibrium scenarios in 1D Bose gases. This is true and GHD is a powerful theory, however there are other analytic and numerical techniques that successfully describe many of these scenarios and have existed well before GHD. Is there a reason this theory has been singled out? Or can this sentence be reworded to clarify its intent?
2 - The authors refer to a naive mean-field result from which their predicted dynamics of the homogenous gas differs at long times. In the main text they find the dynamics scale at longs times as $n(t)\sim t^{-1/2}$ which they say "deviates significantly from the mean-field prediction $n(t)\sim t^{-1}$ obtained from the equation $\partial_{t}n=-\kappa n^{2}$." Can this mean-field result and associated equation be expanded and clarified in terms of where it comes from and how it is determined, either with references to existing work or further explanation in the text?
3 - Can a reasonable explanation (intuitive or otherwise) be given as to why the loss dynamics in the weakly-confined limit end up behaving as in the strongly-confined regime at long times? And why such a crossover might be expected at $T_{\omega}/4$?
4 - The reviewer has noticed a number of grammatical errors throughout the manuscript which should be rectified. Some are pointed out below but the reviewer requests that the authors thoroughly examine their manuscript for a number of others.
Requested changes
A - Check and fix grammatical and typographic errors (which include but are not limited to: a number of missing words, spelling, typos, flow). For example, one mathematical typo is the $\tau$ appearing in the definition of $g(\tilde{t})$. Here, this $g$ no longer refers to the previously defined two-body interaction strength but to a newly introduced function, which is not ideal, using another symbol would be recommended. One grammatical example can be found on page 13, the beginning of the paragraph above Eq (38) should read: "In order [to] present a quantitative theory of the short-time dynamic[s]...".
B- Further clarity and explanation regarding some of the derivations. As an example, the two equations (18) presented at the beginning of Sec 3.1, do these result simply from the new definitions above or were some other mathematical steps or approximations taken to arrive at these equations? Writing more clearly the necessary mathematical steps for some of these derivations would be helpful and bring much more clarity.
C- In Figure 1, the Right Panel, could further explanation be given concerning the difference between the circle/square results and the solid lines. What is the difference between numerically solving Eqs. (17), and taking the numerical results for $\tilde{n}(\tilde{t})$ and inserting them into Eq. (20)? The reviewers misunderstanding on this point likely results from point B above; a misunderstanding of where exactly Eqs. (18) have come from.
D - In Figure 3, as with Figure 5, is there a reason that the label sizes are unnecessarily large compared to the main text and other figures? And the quantity being plotted is $\tilde{n}$, whereas the figure caption reads $\tilde{N}$?
Author: Lorenzo Rosso on 2021-10-11 [id 1835]
(in reply to Report 1 on 2021-07-30)
The authors study the population dynamics of both a homogenous and harmonically confined 1D Bose gas with strong two-body losses using a set of rate equations applicable in the quantum Zeno regime (strong dissipation). In the homogenous case the results extend those derived for a gas on a lattice in Ref [36], as well as those for the continuum situation in Ref [37], and deviate from a naive mean-field prediction in the long-time limit. Their methodology allows them to extend their results to include a trapping potential and show how the gas is depleted in the experimentally relevant scenario of harmonic confinement. In this case, they predict that the depletion is accelerated and that a novel decay regime arrises where the behaviour of the gas is consistent with strong-confinement at long times, even when the gas is only weakly trapped.
Overall the reviewer finds the work worthy to be published, but only after revision of the manuscript to enhance the quality of communication of the results reported in the paper.
A : We thank the reviewer for the careful reading of the manuscript and for supporting publication in SciPost Physics. We believe to have significantly improved the communication of our results thanks to her/his constructive criticism.
Q1: In the introduction, generalised hydrodynamics (GHD) is identified as a theory which successfully describes out-of-equilibrium scenarios in 1D Bose gases. This is true and GHD is a powerful theory, however there are other analytic and numerical techniques that successfully describe many of these scenarios and have existed well before GHD. Is there a reason this theory has been singled out? Or can this sentence be reworded to clarify its intent?
A1: GHD has been highlighted because the theoretical treatment developed in Sec. 4 precisely consists in performing a GHD study, as it was also pointed out by the other reviewer. We have not stressed too much this point in the previous version because our treatment could have been proposed also before the development of GHD simply invoking the local density approximation and several well-known facts about free-fermion dynamics. In order to solve this issue (reviewer 2 is asking us to mention explicitly GHD), we changed the text in the introduction and wrote "...several developments, among which generalised hydrodynamics [7,8], have produced a theoretical framework that..." to highlight that GHD is only one among many theories that can be used to tackle these problems. On the other hand, to clarify the connection with the main text, we have added a sentence in Sec. 4 that reads: "This approach is an example of the treatments based on the so-called generalised hydrodynamics that has been recently introduced to discuss the dynamics of one-dimensional integrable models with Bethe-ansatz techniques."
Q2: The authors refer to a naive mean-field result from which their predicted dynamics of the homogenous gas differs at long times. In the main text they find the dynamics scale at longs times as $n(t) \sim t^{-1/2}$ which they say ``deviates significantly from the mean-field prediction $n(t) \sim t^{-1}$ obtained from the equation $\partial_t n = -k n^2$." Can this mean-field result and associated equation be expanded and clarified in terms of where it comes from and how it is determined, either with references to existing work or further explanation in the text?
A2: The correct equation that describes the population of the homogeneous gas under local two-body losses has been presented in several articles (see for instance [Durr et al. Phys. Rev A 79 023614 (2009)] or [Bouchoule et al. SciPost Phys. 9 044 (2020)]) and reads:
$$ \dot n = - \kappa n^2 g^{(2)} (0), \qquad \text{with} \quad g^{(2)}(0) = \frac{\langle \hat n( \hat n-1)\rangle_t}{\langle \hat n \rangle_t^2}. $$In uncorrelated or weakly-correlated bosonic gases, it is often the case that $g^{(2)}(0)=1$ is a good approximation, and the mean-field behaviour is recovered. The analytical solution of this differential equation is $n(t) = n(0) / \left(1+ \frac{\kappa}{n(0)} t \right)$, which displays a $t^{-1}$ scaling at long times. This equation has been early used to fit the two-body decay constant, see for instance Ref. [Browaeys et al. Eur. Phys. J. D 8 199 (2000)] and Eq. (1) therein. We have expanded the discussion in the text to clarify this point.
Q3: Can a reasonable explanation (intuitive or otherwise) be given as to why the loss dynamics in the weakly-confined limit end up behaving as in the strongly-confined regime at long times? And why such a crossover might be expected at $T_{\omega} / 4$?
A3: There are two intuitive explanations to answer the second question, and we have added the section 4.4 "From weak to strong confinement'' to explain them.
The first is related to the fact that even if the gas is initially in the weak-confinement limit, it will always perform a rotation in phase space, even if on long time scales. For times much shorter than the harmonic-oscillation period $T_\omega$, the profile of the gas is significantly modified by the loss processes that mostly deplete the denser central region. On longer time scales, the rotation starts to act. At time $T_\omega / 4$ the long-lived population at the edges of the trap has moved to the centre of the trap and there it is rapidly dissipated because of losses. Two different behaviours are thus expected, separated by the time $T_\omega /4$.
A second intuitive explanation has been suggested by Reviewer 2, who proposed to look at the comparison between the oscillation frequency and the instantaneous decay time
$$ r(t) = \frac{1}{\omega} \times \left|\frac{1}{\tilde N (\tilde t)}\frac{d \tilde N (\tilde t)}{d t} \right|. $$One can identify $r \gg 1$ as the limit of weak confinement, whereas $r \ll 1$ is the limit of strong confinement. In the paper we show that assuming to be initially in a regime of weak confinement, $r(t)$ is a decreasing function of time that goes to zero, and thus must enter the regime of strong confinement. By setting the threshold time as $t_{tr}=T_{\omega}/4$ we get $r(t_{tr})=\pi$ as $\mathcal{O}(1)$ threshold value for the parameter $r$. This result also provide us with a possible intuitive explanation of the effectively strongly-confined regime emerging at large times. For $r\ll1$ the oscillation frequency imposed by the trap is much larger than the spontaneous decay time. This fast dynamics induces an effective and fast redistribution of the inhomogeneities in the phase-space distribution due to losses events.
Q4: The reviewer has noticed a number of grammatical errors throughout the manuscript which should be rectified. Some are pointed out below but the reviewer requests that the authors thoroughly examine their manuscript for a number of others.
A4: We have carefully reviewed the entire article and rewritten several parts of the introduction and of the conclusions. We thank the reviewer for pointing out the existence of so many mistakes.
Q A: Check and fix grammatical and typographic errors (which include but are not limited to: a number of missing words, spelling, typos, flow). For example, one mathematical typo is the $\tau$ appearing in the definition of $g(\tilde t)$. Here, this $g$ no longer refers to the previously defined two-body interaction strength but to a newly introduced function, which is not ideal, using another symbol would be recommended. One grammatical example can be found on page 13, the beginning of the paragraph above Eq (38) should read: "In order [to] present a quantitative theory of the short-time dynamic[s]...".
A A: We corrected the typos. We have changed the name of the function $g(\tilde t)$ with $\nu (\tilde t)$.
Q B: Further clarity and explanation regarding some of the derivations. As an example, the two equations (18) presented at the beginning of Sec 3.1, do these result simply from the new definitions above or were some other mathematical steps or approximations taken to arrive at these equations? Writing more clearly the necessary mathematical steps for some of these derivations would be helpful and bring much more clarity.
A B: In order to better clarify the mathematical steps behind the derivations of aforementioned equations, we added a new appendix titled "Homogeneous Bose gas: derivations". This material was not included in the main text in order not to make it too long. Furthermore, we made explicit an additional step [see Eq. (23)] in the derivation of the long-time behaviour (Sec. 3.2).
Q C: In Figure 1, the Right Panel, could further explanation be given concerning the difference between the circle/square results and the solid lines. What is the difference between numerically solving Eqs. (17), and taking the numerical results for $\tilde n (\tilde t)$ and inserting them into Eq. (20)? The reviewers misunderstanding on this point likely results from point B above; a misunderstanding of where exactly Eqs. (18) have come from.
A C: The point of plotting the solid lines is to show that $n_{\tilde k}(\tilde t)$ is completely determined by the value of $\tilde n(\tilde t)$ according to the relation:
$$ n_{\tilde k}(\tilde t) = \begin{cases} \sqrt{\tilde n(\tilde t)} e ^{- 2\pi \tilde k^2 \int_{0}^{\tilde t} \tilde n(\tilde t') {\rm d} \tilde t'}, & \tilde k \in[- \pi , \pi ];\\ 0, & \text{otherwise}. \end{cases} $$Since the solid lines obtained from this formula agree with the numerical ones, we have validated this relation.
Q D: In Figure 3, as with Figure 5, is there a reason that the label sizes are unnecessarily large compared to the main text and other figures? And the quantity being plotted is $\tilde n$, whereas the figure caption reads $\tilde N$?
A D: We have decreased the size of the labels in the figures. In Fig. 3 we are plotting $\tilde N (\tilde t)$, which is defined in Eq. (39) of the current version.
Anonymous on 2021-08-05 [id 1634]
I am the author of the report 2. I would like to here to elaborate a bit more on the reason why I suggest that, although this paper is of very good quality, it should be published in Scipost Physics Core and not in Scipost Phycis. I did not think it really meets either of the 4 criteria of Scipost Physics. More precisely, it elaborates on the results of [Phys. Rev. A 103, L060201 (2021)] and for this reason it is not really a " Detail a groundbreaking theoretical/experimental/computational discovery". It discovers a behavior not forseen such that I do not think it " Present a breakthrough on a previously-identified and long-standing research stumbling block". I also doubt it meets the 2 other criteria of acceptance in Scipost Physics.
On the other hand, the fact the paper meets or not the criteria of Scipost Phycics is subjective of course. The paper is correct and presents new results that are not trivial, on a subject which presently attracts a lot of attention: it surely merits publication. I let the editor decide whether the paper should be published in Scipost Physics or not.
Author: Lorenzo Rosso on 2021-10-11 [id 1837]
(in reply to Anonymous Comment on 2021-08-05 [id 1634])We thank again the reviewer for this comment. We have answered about this in the reply to the report 2.
Sincerely,
The authors

---

## Round 1 · Referee Report · Anonymous · 2021-8-4

Strengths
1. Very clear
2. Timely
3. Experimentally relevant
Report
In this paper, the authors investigate the case of strong two-body losses in a 1D gas in the continuum. Those results extend those published in Rossini et al. (2020) that concerned gases confined in a lattice. The situation considered is that of strong losses that promote the gas to the fermionized regime and the gas is described by its rapidity distribution (distribution of the momenta of the fermions). Losses, although strongly reduced by Zeno effect, are however not strictly vanishing which leads to an evolution of the rapidity distribution that the authors compute. In this paper, taking the limit of vanishing lattice spacing, the authors derive similar results valid for homogeneous gases in the continuum (i.e., in absence of any potential). The second part of the paper is devoted to the effect of a longitudinal harmonic potential. Using Local density approximation, equivalent to the Generalized HydroDynamics approximation, they compute the time evolution of the system.
The paper is well written and very clear. The results are new, interesting and relevant for experimental situations. I strongly support publication. However, I am not sure it meet the criteria for Scipost Physics. It might be more appropriate for Scipost Physics Core.
I have moreover several comments, that the authours might consider.
1. In the introduction, when citing works about the cooling induced by losses, I would cite [SciPost Phys. 8, 060 (2020) ] which is relevant I think: this paper solve the question about the asymptotic temperature imposed by losses, question raised by the paper [Rauer2016,PRL116,030402].
2. When introducing the Lindbladian in Eq. 1a,1b, the authors might mention the fact that the underlying markovian approximation (reservoir of infinite width)
make sens only in 1D (see ref [20])
3.When discussing the fact that the density matrix Eq. 7 is a stable mode of eq.1, I would present the result slightly differently. I would recast Eq. 1 as
$\partial \rho/\partial t = 1/\hbar (-i H_{NHLL} \rho + i \rho H_{NHLL}^+ ) + \gamma/2 \int \psi^2(x) \rho \psi^{+2}(x) $. Because, within the fermionized approximation, $H_{NHLL}$, given by Eq. 5, is hermitian, and because $\rho$ given in eq 6 commutes with $H_{NHLL}$, the first term of the right hand side is vanishing. The second term is vanishing because $\psi^2(x)$ is vanishing for fermionized states.
4. At the end of section 2.1, I have a comment about the "standard time-of-flight". People use this term also in 3D gases. However in this case, what is measured is in general very different from the initial bosonic momentum distribution since the width of the far field velocity distribution is dominated by the initial interaction energy. It is only in reduced dimension, because of the fast transverse expansion that amounts to an effective instantaneous switch off of interaction at the beginning of the time-of-flight, that one measures the bosonic momentum distribution with a "standard time-of-flight". I would put a sentence to make sure there is no ambiguity.
5. After eq. 15, I do not like the sentence "accordingly $n_k$ is an adimensional quantity".$ n_k$ is defined properly already. We now it is adimensionnal. We do not learn this from Eq. 15
6. Before Eq. 16 : "the initial momentum profile, $n_k(0)$" might be misleading. We are not considering the momentum profile of the bosons, but that of the equivalent fermions, which is very different. Why not using the word "rapidity" to avoid any confusion ? Otherwise, specify "fermionic"
7. After eq.20, I would replaced "the variance of the momentum distribution" by "the variance of the Gaussian" (only in the long time limit, both coincide.
8. Introducing the Maxwell-Boltzman distribution make sense only if $|\mu|\gg k_B T$. The authors should mention that this is indeed the case at large time.
9. In section 3.2, when introducing $\tilde g$, a misprint appears : $\tau$ should be replaced by $\tilde t$.
10. In section 3.3, the parameters of the numerical calculation are given are not very instructive. The authors might say something like "the numerical calculation has converged within x% for dt= ...."
11. At the beginning of section 4, I would use the word Generalized HydroDynamics . It is true that the GHD approximation is nothing else than the LDA approximation, together with the fact that the local equilibrium is described by the (local) rapidity distribution and not only simply by \mu and T. The term LDA has been used since a long time in the cold atom community. However, one should acknowledge the important breakthrough made recently by the people from the Bethe-Ansatz community, that introduce the term GHD instead of LDA.
12. When introducing Eq.36, I think the sentence is not very clear. I would say " We introduce $\tilde N$ defined by $\tilde N = N/N_{initial}$."Since the area of the initial circle is $pi$, we have ....
13. In eq.34, I think there is a misprint : $\tilde \Gamma=\Gamma_{eff}\pi^3n_{in}(0)^3$
13. I would discuss differently the different regimes of the behavior of $\tilde n$ in the harmonic case. I would separate the different regimes by the ratio $r=(\dot{n}/n) /\omega $. Here $\dot{n}/n$ is the typical $\dot{n}/n$ due to losses, computed at the center of the cloud. Two regimes occurs : $r\ll 1$ and $r\gg 1$.
If the condition $r \ll 1$ is fulfilled, then the discussion of section 4.4 holds and one expects the behavior given in eq. 39. For initial situation such that $\omega /(\Gamma_{eff} n_{ini}(0)^3) \gg 1 $ the condition $r\ll 1$ is verified from the beginning.
If on the other hand, $1/(\Gamma_{eff}n_{in}(0)^3) \ll 1/\omega$, then this regime is attained only for $t\gg 1/\omega$ (see below).
Let us consider now the condition $r\gg 1 $.
We consider only times large compared to $1/(\Gamma_{eff}n_{in}^3)$, such that, under the effect of losses, $\dot{n}/n \simeq -1/(2t) $.
The condition $r\ll 1$ then reduces to $\omega t \ll 1$. Then Eq. (38) holds. Such a regime occurs only if $1/(\Gamma_{eff}n_{in}(0)^3) \ll 1/\omega$. (ie $\omega/\tilde\Gamma \ll 1$).
This point of view is a suggestion. It is not requiered that the authors change their presentation.
14.In Fig 3, I would put as vertical dotted lines the lines corresponding to the condition $\omega t =1$ (or $\pi/2$), with the appropriate colors. This way one could see that the transition from the regime $r\gg1$ and the regime $r \ll 1$
occurs indeed for the condition $\omega t \simeq 1$.
15. Two minor points concerning the conclusion.
Why the authors say that "the fact that we are working in the strongly dissipative Zeno regime makes this [the population of excited transverse modes] not implausible." ? It is not clear.
Also, where would "interaction effects among the fermionic momenta" come from ?
Requested changes
Among the comments given in the above report, points 6,7 and 9 are requested changes.
Author: Lorenzo Rosso on 2021-10-11 [id 1836]
(in reply to Report 2 on 2021-08-04)
In this paper, the authors investigate the case of strong two-body losses in a 1D gas in the continuum. Those results extend those published in Rossini et al. (2020) that concerned gases confined in a lattice. The situation considered is that of strong losses that promote the gas to the fermionized regime and the gas is described by its rapidity distribution (distribution of the momenta of the fermions). Losses, although strongly reduced by Zeno effect, are however not strictly vanishing which leads to an evolution of the rapidity distribution that the authors compute. In this paper, taking the limit of vanishing lattice spacing, the authors derive similar results valid for homogeneous gases in the continuum (i.e., in absence of any potential). The second part of the paper is devoted to the effect of a longitudinal harmonic potential. Using Local density approximation, equivalent to the Generalized HydroDynamics approximation, they compute the time evolution of the system. The paper is well written and very clear. The results are new, interesting and relevant for experimental situations. I strongly support publication.
A: We thank the referee for the careful reading of the manuscript, for her/his constructive criticism, and for the words of praise.
Q: However, I am not sure it meet the criteria for Scipost Physics. It might be more appropriate for Scipost Physics Core.
The reviewer elaborate on this subject in a later comment:
I am the author of the report 2. I would like to here to elaborate a bit more on the reason why I suggest that, although this paper is of very good quality, it should be published in Scipost Physics Core and not in Scipost Phycis. I did not think it really meets either of the 4 criteria of Scipost Physics. More precisely, it elaborates on the results of [Phys. Rev. A 103, L060201 (2021)] and for this reason it is not really a "Detail a groundbreaking theoretical/experimental/computational discovery". It discovers a behavior not forseen such that I do not think it "Present a breakthrough on a previously-identified and long-standing research stumbling block". I also doubt it meets the 2 other criteria of acceptance in Scipost Physics. On the other hand, the fact the paper meets or not the criteria of Scipost Phycics is subjective of course. The paper is correct and presents new results that are not trivial, on a subject which presently attracts a lot of attention: it surely merits publication. I let the editor decide whether the paper should be published in Scipost Physics or not.
A: We thank the reviewer for this comment. When we started working on this article, one of us had just finished an article on two-body losses for a Bose gas in a lattice [Rossini et al. Phys. Rev. A 103 L060201 (2021)]. The discussion on whether a trap accelerates or decelerates the loss mechanism was debated among the authors of that article, which decided to mention the point in the conclusions. We decided to attack the problem, and we thought that it would be most interesting to do it for a gas in the continuum because: (i) it would have shown us the generality of the previous results, potentially enlarging the impact of our study on the experimental community; and (ii) it would have been a more natural setting for carrying out a local-density approximation analysis. We would have agreed with the reviewer if we had written a paper only on point (i). Yet, the new article considerably extends the previous one including also the effect of the trap. To our knowledge, this is the first time that a lossy and correlated gas in an inhomogeneous setting is characterised, and we are the first to point out that the trap can induce a qualitatively novel decay behaviour (not only quantitative, as we were actually expecting). The result is fundamental for comparing with any experiment: the theory for the homogeneous gas simply does not work if one has an important trap (or waits for long-enough times). Moreover, it develops a connection with generalised hydrodynamics, a theory that has shown its potential exactly in the treatment of gases in "strange'' confining potentials, and the authors of the groundbreaking paper [Schemmer et al. Phys. Rev. Lett. 122 090601 (2019)] mention explicitly that atom losses are a problem for developing a quantitative description of the long-time behaviour of the gas. Although we here deal with a different setting, we are doing the first important step in the right direction (again, nobody has ever characterised the interplay between an external potential and losses). For these reasons, we believe that our article adds a significant new piece of information to the theory of losses in strongly correlated quantum gases and we ask the reviewer and the editor to consider the article for publication in SciPost Physics, as it complies with both criteria 1 and 2. We think that a possible source of confusion could have been the fact that much of our discussion on the harmonic confinement is actually put in the appendix "Rate equation in the strong confinement limit''. The reason for this choice was that we did not want to make the main text too long. Yet, after reading the remark by the referee, and given the large amount of novel material that it contains, we have decided to promote it to the main text as a new subsection. To further stress our point, we are proposing a new title to the article: The one-dimensional Bose gas with strong two-body losses: the effect of the harmonic confinement.
Q1: In the introduction, when citing works about the cooling induced by losses, I would cite [SciPost Phys. 8, 060 (2020) ] which is relevant I think: this paper solve the question about the asymptotic temperature imposed by losses, question raised by the paper [Rauer2016,PRL 116,030402].
A1: We thank the referee, the paper is cited in the present version (in fact, it did not appear in the previous list of references because of a broken bibtex link).
Q2: When introducing the Lindbladian in Eq. 1a,1b, the authors might mention the fact that the underlying markovian approximation (reservoir of infinite width) make sens only in 1D (see Ref. [20])
A2: We mentioned this fact by adding the sentence "which makes sense only in 1D (see Ref. [20])" before introducing the Lindbladian in Eq. 1a, 1b.
Q3: When discussing the fact that the density matrix Eq. 7 is a stable mode of eq.1, I would present the result slightly differently. I would recast Eq. 1 as $\partial \rho / \partial t = 1/ \hbar ( -i H_{NHLL} \rho + i \rho H_{NHLL}^{\dagger}) + \gamma / 2 \int \psi(x)^2 \rho \psi(x)^{\dagger 2}$ . Because, within the fermionized approximation, $H_{NHLL}$ , given by Eq. 5, is hermitian, and because $\rho$ given in eq 6 commutes with $H_{NHHL}$, the first term of the right hand side is vanishing. The second term is vanishing because $\psi(x)^2$ is vanishing for fermionized states.
A3: We have rewritten the paragraph and presented the discussion as suggested. The new paragraph is in SubSec. 2.1 and starts with "In order to prove that the latter density matrix is a stable mode of the master..."
Q4: At the end of section 2.1, I have a comment about the "standard time-of-flight". People use this term also in 3D gases. However in this case, what is measured is in general very different from the initial bosonic momentum distribution since the width of the far field velocity distribution is dominated by the initial interaction energy. It is only in reduced dimension, because of the fast transverse expansion that amounts to an effective instantaneous switch off of interaction at the beginning of the time-of-flight, that one measures the bosonic momentum distribution with a "standard time-of-flight". I would put a sentence to make sure there is no ambiguity.
A4: We thank the referee for the remark: in the main text we have clarified explicitly that we are referring to a time-of-flight experiment on a one-dimensional bosonic gas.
Q5: After Eq. 15, I do not like the sentence "accordingly $n_k$ is an adimensional quantity". $n_k$ is defined properly already. We know it is adimensionnal. We do not learn this from Eq. 15
A5: We changed the sentence from "accordingly $n_k$ is an adimensional quantity" to "since $n_k$ is an adimensional quantity" in order to avoid misunderstanding.
Q6: Before Eq. 16 : "the initial momentum profile, $n_k(0)$" might be misleading. We are not considering the momentum profile of the bosons, but that of the equivalent fermions, which is very different. Why not using the word "rapidity" to avoid any confusion ? Otherwise, specify "fermionic"
A6: We write "the initial fermionic momentum profile, $n_k(0)$''; we prefer avoiding the use of the word "rapidities'' because we are not using it systematically in the text. We also think that this writing could make the article more obscure to experimentalists and theorists not familiar with the Bethe-Ansatz theory.
Q7: After Eq. 20, I would replaced "the variance of the momentum distribution" by "the variance of the Gaussian" (only in the long time limit, both coincide).
A7: We modified the text accordingly.
Q8: Introducing the Maxwell-Boltzman distribution make sense only if $\mu \gg k_B T$. The authors should mention that this is indeed the case at large time.
A8: The inequality written by the reviewer states under which condition the Bose-Einstein and Fermi-Dirac quantum statistics reduce to a Maxwell-Boltzmann classical statistics. In principle, we are studying an out-of-equilibrium protocol, and nothing tells us that we should have a fermionic momentum distribution function that has some resemblance with well-known equilibrium ones. This is the case, for instance, at intermediate times, where we have identified a truncated Gaussian. However, we are very grateful to the reviewer for this beautiful observation concerning the distribution at long times, that further reinforces an interpretation in terms of a quantum gas that has become classical. We added a sentence on this point as requested by the referee.
Q9: In section 3.2, when introducing $\tilde g$, a misprint appears :$\tau$ should be replaced by $\tilde t$.
A9: We thank the referee for spotting the mistake. We corrected it.
Q10: In section 3.3, the parameters of the numerical calculation are given are not very instructive. The authors might say something like "the numerical calculation has converged within x$\%$ for $dt=$ ...."
A10: We believe that making the parameters of the numerical simulations explicit is useful for the more numerically-oriented reader, and could help the reproduction of the data. Since it is just the matter of few lines, we prefer to leave the article as it is. However, as requested by the referee, we added a comment about the convergence of the numerical integration for this choice of parameters.
Q11: At the beginning of section 4, I would use the word Generalized HydroDynamics. It is true that the GHD approximation is nothing else than the LDA approximation, together with the fact that the local equilibrium is described by the (local) rapidity distribution and not only simply by $\mu$ and $T$. The term LDA has been used since a long time in the cold atom community. However, one should acknowledge the important breakthrough made recently by the people from the Bethe-Ansatz community, that introduce the term GHD instead of LDA.
A11: The novel version of the article mentions explicitly the generalised hydrodynamics in the section 4; its importance is moreover highlighted in the first paragraph of the introduction, where it is the only technique that we explicitly mention among a number of theoretical schemes that have been discussed over the years for treating the dynamics of one-dimensional (Bose) gases.
Q12: When introducing Eq.36, I think the sentence is not very clear. I would say " We introduce $\tilde N$ defined by $\tilde N = N/ N_{initial}$. "Since the area of the initial circle is $\pi$, we have ...."
A12: We changed the sentence into "We will study the time evolution of the rescaled number of atoms $\tilde N$, defined by $\tilde N = N / N(0)$. Since the area of the initial circle is $\pi$, we have:" in order to clarify this point.
Q13a: In Eq.34, I think there is a misprint : $\tilde \Gamma = \Gamma_{eff} \pi^3 n_{in}(0)^3$.
A13a: We thank the referee for spotting the mistake. We corrected it.
Q13b: I would discuss differently the different regimes of the behavior of $\tilde n$ in the harmonic case. I would separate the different regimes by the ratio $r = (\dot n /n)/\omega$. Here $\dot n / n$ is the typical $\dot n / n$ due to losses, computed at the center of the cloud. Two regimes occurs : $r \ll 1$ and $r \gg 1$.
A13b: We appreciated the suggestion of the reviewer, and since we liked her/his argument, we have inserted it in the article. It appears in the subsection 4.4 "From weak to strong confinement'', where it is used to argue that the transition from weak confinement to strong confinement should take place at $\omega t_{tr} = \pi/2$ (i.e. $t_{tr}=T_{\omega}/4$), corresponding to $r(t_{tr})=\pi$. The contribution of the referee is acknowledged at the end of the article.
Q14: In Fig 3, I would put as vertical dotted lines the lines corresponding to the condition $\omega t = 1$ (or $\pi/2$), with the appropriate colors. This way one could see that the transition from the regime $r \gg 1$ and the regime $r \ll 1$ occurs indeed for the condition $\omega t \simeq 1$.
A14: We see the point raised by the reviewer, and appreciate very much his suggestion. Since the plot is already rather full of curves we think that implementing it for every value of $\omega / \tilde \Gamma$ would make the figure too chaotic. We have thus added a dashed vertical line signaling $t_{tr}=T_{\omega}/4$ only for $\omega / \tilde \Gamma = 0.05$, $0.1$ and $0.2$, since are the curves that better show the transition between the two regimes.
Q15: Two minor points concerning the conclusion. Why the authors say that "the fact that we are working in the strongly dissipative Zeno regime makes this [the population of excited transverse modes] not implausible." ? It is not clear. Also, where would "interaction effects among the fermionic momenta" come from ?
A15: A one-dimensional gas is typically obtained by confining the gas in two transverse directions with a strong potential that can be approximated by a two-dimensional harmonic oscillator (HO). We speak of a one-dimensional system when all particles are confined to the ground state of this two-dimensional HO; in general one simply checks that $\hbar \omega \gg k_B T$. The problem arises when one considers strong dissipation or strong interactions and performs an experiment that extends over a long time scale (compared to the typical time of the kinetic energy). In these cases, discussions with experimentalists tell us that scattering events mediated for instance by elastic collisions, can populate the higher bands in a non-negligible way. Similar comments can also be found in published articles, see for instance the Sec. V in Ref. [Mark et al. Phys. Rev. Res. 2 043050 (2020)] or just the abstract of Ref. [Zhu et al. Phys. Rev. Lett. 112 070404 (2014)], although in these articles they speak of the bands of an optical lattice, and not of the transverse confinement. To the best of our knowledge, a systematic experimental characterisation of these processes has not yet been performed, although it would be highly desirable. We could mention Ref. [Buchler, Phys. Rev. Lett. 104 090402 (2010)] as a theoretical example where the effect of higher bands in the context of elastic scattering is studied. Coming back to our dissipative problem, the theoretical study has been often blocked by the exceptional difficulty of the numerical analysis: even the simulation of the purely one-dimensional gas is challenging. However, our rate equations are very simple and not-at-all demanding from the numerical viewpoint. The inclusion of higher bands in an effective way could be easily done. We are trying to actively develop connections with experimentalists in order to be better guided in the development of the most appropriate theory. One final comment concerning the interaction among fermionic momenta. We are studying a gas that has strong elastic and inelastic two-body losses. If elastic and inelastic two-body losses would be infinite, no dynamics would take place: the gas would simply fermionise. Our work takes into account the effect of a large but finite dissipation rate. It would be interesting to study also the effect of a large but finite elastic collision. We know by general arguments (see for instance Ref. [Lange et al. Phys. Rev. B 97 165138 (2018)]) that the Hamiltonian generates a less important correction (the Fermi golden rule tells has that an Hamiltonian with strength $\lambda$ scatters population among different energy eigenstate with a rate scaling as $\lambda^2$). Yet, its effect will be present, and could affect the dynamics in phase space in interesting ways, e.g. with diffusion processes (but at this stage this is completely speculative).

---

## Round 2 · Referee Report · Anonymous (Referee 1) · 2021-11-25

Report

The reviewer thanks the authors for their time taken to improve the manuscript. The current manuscript reads well and the included additions have created a thorough and well crafted explanation of their timely and useful results.

The reviewers recommendation is to publish after correcting two small typos:

  1. In Figure 3 the black and cyan dashed lines should be labeled as Eq. (42) and Eq. (45), rather than the current labels of Eq. (40) and Eq. (43).

  2. In Appendix C the text before Eq. (69) should read "We can also present an analytical formula for ν(˜t)", rather than "... also for g(˜t)".

---

## Round 2 · Referee Report · Anonymous (Referee 2) · 2021-12-16

Report

I thank the authors for their detail answers to all the comments and questions of the first referee report. I agree with their answers and I think the paper is now ready for publication. I agree it can be publiched in Scipost Physics.

---

## Round 2 · List of Changes

(1) We propose a new title to the article "The one-dimensional Bose gas with strong two-body losses: the effect of the harmonic confinement", in order to further stress the fact that our work extends the previous one (i.e. Ref.[36]) including also the effect of the trap.

(2) Two important remarks have been added in the Introduction. The former deals with highlighting the fact that GHD is only one among many theories that can be used to tackle these problems (see the sentence "...among several developments..."). The latter remark points out the two different regimes of confinement due to the presence of the trap, namely weak and strong confinement (see the part "Two regimes are identified: ...").

(3) Thanks to the observation of the reviewer 2 we have rewritten the paragraph in SubSec. 2.1 starting with "In order to prove that that the latter density matrix..."

(4) Thanks to a remark of the reviewer 1, in Sec. 3.2 we changed the name of the function g (\tilde t) with \nu (\tilde t)

(5) Thanks to a remark of the reviewer 1, in Sec. 3.2 we enlarged the discussion concerning the mean-field limit. See the paragraph starting "We find a long-time behaviour characterised by n(t)..."

(6) In Sec. 3.2 we added an equation, namely Eq. (23) in order to better clarify the mathematical steps in the derivation.

(7) We modified Fig. 3 including three vertical dashed lines marking the crossover between weak- and strong-confiniment. Moreover, the dimension of the labels has been reduced as requested.

(8) We added a new figure, namely Fig. 4, in which the spatial density profile \tilde N( \tilde x, \tilde t) is plotted.

(9) Thanks to a suggestion of the reviewer 1, the discussion in Sec. 4.4 has been enlarged. In particular, a new adimensional parameter r(t) has been introduced to mark the switch from weak- to strong-confinement. The contribution of the reviewer has been acknowledged.

(10) We modified the dimension of the labels of Fig. 5 (Fig. 4 of the previous version) as requested.

(11) Sec. 4.5 and Sec. 4.6 have been introduced in the main text. In the previous version of the paper they were in the Appendix E. We decided to move it to the main text in order to emphasize the discussion about the harmonic confinement.

(12) We added a new Appendix (now Appendix. A) titled "Homogeneous Bose gas: derivations" in order to better clarify the mathematical steps to derive Eqs. 19 and 21.

(13) We modified the dimension of the labels of Fig. 6 (Fig. 5 of the previous version) as requested.

(14) We changed the title of Appendix. E into "Analytical solution for \omega / \tilde \Gamma = 0"

---

## Editorial Decision

published